# Scaling Channel-Invariant Self-Supervised Learning for Microscopy Images

## Abstract

Recent advances in self-supervised pre-training of foundation models for natural images have made them a popular choice for various visual systems and applications. Self-supervised strategies have also shown promise in non-RGB scientific imaging domains such as in biology, medical and satellite imagery, but their broader application is hampered by heterogeneity in channel composition and semantics between relevant datasets: two datasets may contain different numbers of channels, and these may reveal distinct aspects of an object or scene. Recent works on channel-invariant strategies report substantial advantages for those that account for variable channel compositions without sacrificing the ability to jointly encode channels; yet, how these strategies behave at scale remains unclear. We here show that, surprisingly, trained across large-scale microscopy datasets, independent-encoding of channels consistently outperforms joint-encoding methods by a substantial margin. We explore this result along an extensive set of experiments and open-source a new general purpose feature extractor for fluorescent microscopy images, DINO BoC, that sets a new state-of-the-art across challenging benchmarks, including generalization to out-of-distribution tasks and unseen channel combinations at test time.

## 1 Introduction

By enabling scientists to reveal the substructural composition and dynamics of cells and tissues, fluorescent microscopy has enabled countless scientific discoveries that collectively underpin modern medicine and our understanding of life. Owing to a recent confluence of effective protocols to reveal distinct subcellular structures across multiple channels, and to automate image acquisition across hundreds of experimental conditions and thousands of cells, high-content fluorescent microscopy is emerging as a powerful platform to uncover mechanisms of disease, to accelerate drug discovery, and to interrogate cellular biology at unprecedented scale and salient detail (Chandrasekaran et al., 2021). In addition, vast amounts of microscopy data that have been generated over the last decades wait to be leveraged to train general purpose foundation models to comprehensively represent the morphological "body-language" of cells.

Towards this goal, we here focus on a key technical challenge that, along with many other scientific imaging domains (Zhu et al., 2022), distinguishes fluorescent microscopy data from natural images: to observe specific biological phenomena of interest, biologists routinely design bespoke imaging protocols to reveal distinct sets of cellular structures. The data-landscape of fluorescent microscopy thus consists of a vast number of small-to-moderate scale datasets that vary both in the number of channels they contain, and with respect to the biological semantics of each channel (see Fig. 1). In contrast to natural (RGB) images, most models trained on one fluorescent microscopy dataset can thus neither be re-used in other studies, nor draw on other datasets, yielding representations of limited expressivity and that generalize poorly (Chen et al., 2024). Learning powerful unified feature extraction models for scientific imaging domains with variable channel composition therefore requires channel-invariant methods.

Fluorescent microscopy has proven a particularly fruitful test-bed for the development of such methods. As proposed by (Xun et al., 2024), a technically trivial solution to the challenge of variable channel inputs is to simply abandon the joint-encoding of channels, pass channels through the model one-by-one and concatenate their output embeddings. This has the theoretical advantage of yielding

truly channel-agnostic models, but sacrifices the ability to (explicitly) learn inter-channel interactions and would thus be expected to broadly fail at key analytical use cases, such as the analysis of protein-localization changes relative to reference channels that provide ground-truth for organelle position (Human Protein Atlas, 2019; Lacoste et al., 2024). Consistently, recent works report significant advantages for more technically sophisticated approaches, that reconcile joint-channel-encoding with variable number of input channels (up to some maximum) through customizations of vision transformers (ViTs) (Bao et al., 2024; Bourriez et al., 2024; Pham & Plummer, 2024).

However, all previous studies employed inconsistent sets of relatively small pre-training datasets, employed supervised learning objectives (Bao et al., 2024), or did not evaluate generalization to out-of-distribution (OOD) tasks and datasets with unseen channel combinations - key metrics of success for the utility of general purpose, channel-invariant feature extractors. We here conduct a first, large-scale study into the scaling properties of channel-invariant methods, across uniform model architectures and learning objectives, and rigorous benchmarks. Completely inconsistent and often missing labels across microscopy datasets render supervised methods ill-suited to this end. We hence base our study on state-of-the-art (SOTA) self-supervised learning (SSL) strategies which have been shown to yield rich representations of cellular morphology on channel-homogeneous microscopy datasets (Doron et al., 2023; Kraus et al., 2024).

In marked contrast to previous results, we find that, at scale, independent channel-encoders, i.e. Bag of Channel (BoC) models, leveraging ViT architectures trained with DINOv2 (Oquab et al., 2023), significantly and consistently outperform joint-channel-encoding methods across an extensive set of testing regimes. Our results pose a broad challenge to the assumption that joint-channel-encoding is beneficial in non-RGB domains. Our main contributions are as follows:

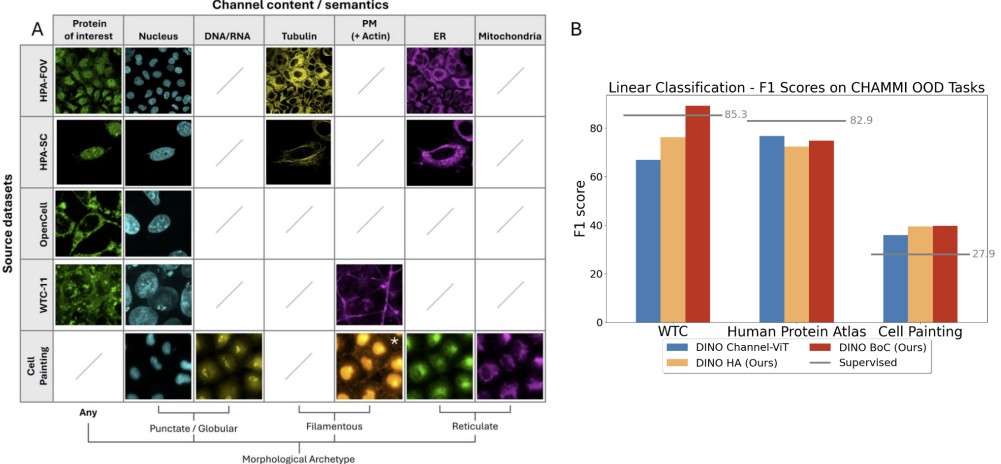

Figure 1: **Training across fluorescent microscopy datasets.** (A) We consolidate images from Human Protein Atlas (2019)(HPA-FOV), Le et al. (2022)(HPA-SC), Cho et al. (2022)(OpenCell), Viana et al. (2023)(WTC) and Doron et al. (2023)(Cell Painting), into a dataset (ExtendedCHAMMI) that reflects the diversity in channel number, order, and semantics, that characterizes fluorescent microscopy studies. Despite some conventions (e.g. the nucleus is usually imaged using blue fluorescing stains), there is no necessary correspondence between specific channels and/or wavelengths, and their biological semantics. Images are pseudo-colored according to emission wavelength. CellPainting visualizes Plasma Membrane (PM) and Actin in one channel. (B) We benchmark self-supervised channel-invariant strategies on their capacity to yield features that generalize to OOD tasks. Our DINO BoC approach compares favorably.

- We conduct an extensive study of channel-invariant SSL methods across large and diverse microscopy datasets.
- Against all previous evidence, we report that independent-encoding of channels outperforms joint-channel-encoding strategies across an extensive set of experiments including in-domain, cross-dataset, and OOD generalization setups, challenging key theoretical assumptions on the optimality of joint-channel-encoding in non-RGB domains.

- We substantiate our results through a set of control experiments to directly analyze the impact of joint versus independent-channel-encoding by virtue of a novel channel-invariant Hierarchical Attention scheme, as well as an ablation of SSL objectives.

- We open-source a new general-purpose feature extractor, DINO BoC, that sets a new SOTA for channel-invariant learning for microscopy.

## 2 RELATED WORK

**Vision transformers and self-supervised learning.**    The goal of self-supervised learning (SSL) is to learn to project the data onto an embedding space such that the features retain the information contained in the original data while being organized in a way that reflects high-level relationships between the data points. In SSL this is achieved by using the samples themselves as the source of supervision, without leveraging additional labels. Learning task-agnostic representations has become pervasive both in Natural Language Processing (Devlin et al., 2019; Radford et al., 2019; Touvron et al., 2023) and, more recently, in Computer Vision (Chen et al., 2020; Caron et al., 2021; He et al., 2022; Assran et al., 2023). The promise of this approach is that it enables the use of large amounts of unlabeled data to learn multi-purpose features that can be applied off the shelf, without fine-tuning, to any downstream tasks, often surpassing the performance of task-specific models. A widely adopted approach involves the use of a contrastive objective, such as in DINOv2 (Oquab et al., 2023) – currently the state-of-the-art in SSL models for computer vision. In contrast, generative models, such as Masked Autoencoders (MAE) (He et al., 2022), are trained by reconstructing masked or corrupted regions of the input.

**Applications of SSL to microscopy images.**    The literature on self-supervised learning for cellular microscopy focuses mainly on models designed for specific datasets (Doron et al., 2023; Kobayashi et al., 2022) or imaging protocols, such as the Cell Painting assay (Kim et al., 2023). However, these pre-trained models cannot be reused across studies with different microscopy configurations. Further, this approach is not viable for learning powerful feature representations for small-scale datasets.

To overcome this limitation, CytoImageNet (Hua et al., 2021) proposed collapsing channels into one by averaging across them. This approach loses the semantic information carried by distinct channels. Alternatively, Microsnoop (Xun et al., 2024) proposed to encode each channel individually with a U-Net (Ronneberger et al., 2015) trained with a masked SSL strategy, and reassembling whole-image representations post-hoc by concatenating the embeddings for each channel.

In contrast, ChAda-ViT (Bourriez et al., 2024), Channel-ViT (Bao et al., 2024), Kraus et al. (2024) and (Pham & Plummer, 2024) studied joint-channel-encoding with ViTs by converting the variable number of channels problem into a variable sequence length problem, as such, channel interactions can be readily learned. Channel-ViT (Bao et al., 2024), in particular, tackles a problem distinct to channel-invariant learning, they focus rather on robustness to missing channels.    Supervised frameworks, such as those of Hua et al. (2021), Bao et al. (2024) and Pham & Plummer (2024) do not fully align with the end goal of channel-invariant models, which aim to be transferable across new microscopy studies and thus require features that are task-agnostic. We here pioneer independent-channel-encoding with ViT backbones and SSL objectives, and benchmark it against joint-encoding (Channel-ViT) methods, while scaling both model and dataset sizes.

## 3 METHOD

### 3.1 INDEPENDENT-CHANNEL-ENCODING

Given a channel-heterogeneous dataset $\mathbb{X}$, where a sample $x^{(j)}$ has $K_j$ channels, let $K_{\max}$ denote the maximum number of channels across all images. The goal of channel-invariant learning is to propose a model that is able to accommodate images with a variable number of channels.

A straightforward strategy to deal with channel number variability is to separately encode each channel, using a common backbone. The individual features obtained for each channel can then be aggregated, e.g., via concatenation, to obtain the image-level representation.

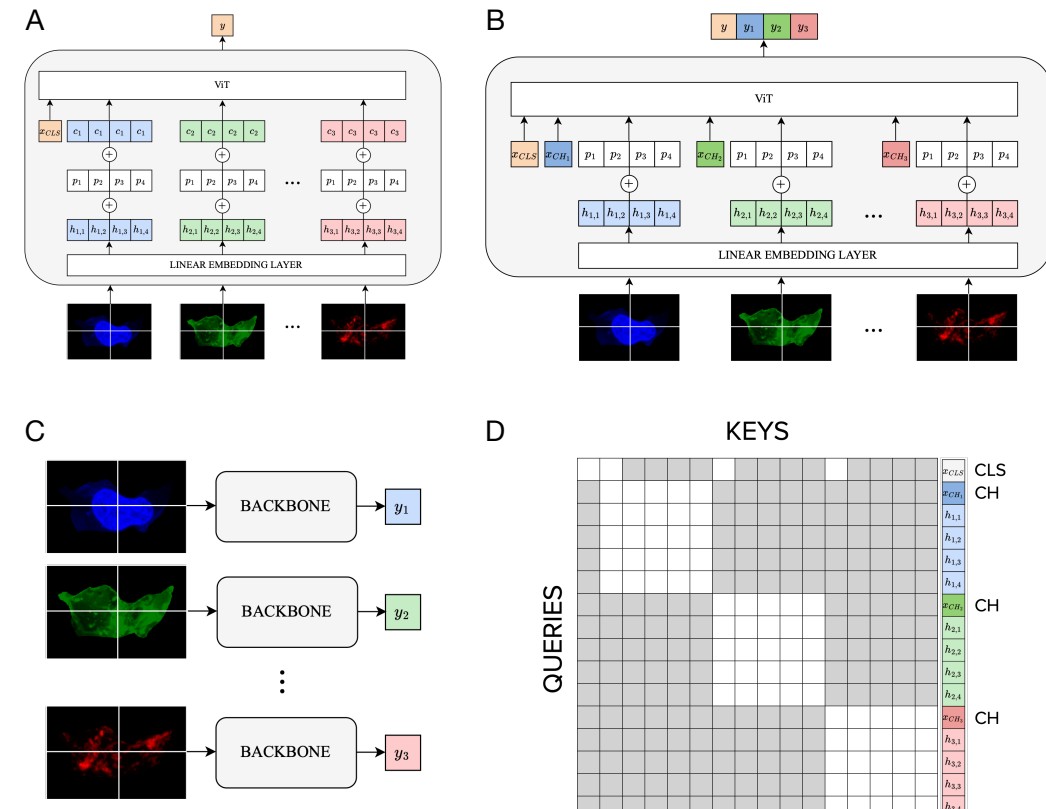

Figure 2: Overview of different channel-invariant strategies. (A) Joint-channel-encoding strategy of Channel-ViT (Bao et al., 2024; Bourriez et al., 2024): the image is reshaped into a sequence of single-channel patches and channel embeddings are used to retain channel information. (B) Hierarchical Attention model: a specialized attention mask is used to enforce independent-channel-encoding via channel class tokens $x_{CH}$, while the global class token $x_{CLS}$ supports inter-channel reasoning. (C) Independent-channel-encoding strategy, note that a common backbone is used. (D) Attention mask of the Hierarchical Attention model.

This is the strategy employed by Microsnoop (Xun et al., 2024), a tool for profiling heterogeneous microscopy images based on a convolutional U-Net (Ronneberger et al., 2015) backbone and trained with a masked SSL strategy.

In this work we pioneer independent channel modeling using ViTs and the DINOv2 SSL framework. We denominate our approach the *DINO Bag of Channels* model (DINO BoC). We demonstrate its superior performance compared to Microsnoop, as well as the advantages of DINOv2 over alternative SSL frameworks. Furthermore, we benchmark it against channel-adaptive methods, that model the channels jointly, disproving the claimed benefits of incorporating inter-channel attention.

## 3.2 JOINT-CHANNEL-ENCODING

The challenge of joint-channel-encoding on datasets with variable number of channels has been addressed by ChAda-ViT (Bourriez et al., 2024) and Channel-ViT (Bao et al., 2024), specifically for transformer architectures. In particular, they propose an adaptation of the patchfication process, and introduce the concept of channel embeddings.

**Multi-channel patch model.** In the original ViT architecture proposed by Dosovitskiy et al. (2021), given an input image $x \in \mathbb{R}^{K \times H \times W}$ and a patch size $S$, the image is reshaped into a sequence of flattened multi-channel patches:

$$x = [x_1 \quad x_2 \quad \cdots \quad x_N], \quad x_i \in \mathbb{R}^{KS^2},$$

where $K$ is the number of channels and $N = HW/S^2$. A linear embedding layer is applied to each patch $x_i$ resulting in a sequence of patch embeddings $h_i$ of dimension $D$, to which a learnable class token $x_{\text{CLS}}$ is prepended:

$$h = [x_{\text{CLS}} \quad h_1 \quad \cdots \quad h_N] \in \mathbb{R}^{D \times (1+N)},$$

To retain positional information, learnable position embeddings are added to the patch embeddings.

**Single-channel patch model.**    To accommodate variable numbers of channels, Bourriez et al. (2024) and Bao et al. (2024) reshape an image into a sequence of flattened single-channel patches:

$$x = [x_{1,1} \quad \cdots \quad x_{1,N} \quad \cdots \quad x_{K,1} \quad \cdots \quad x_{K,N}], \quad x_{k,i} \in \mathbb{R}^{S^2},$$

where, in $x_{k,i}$, $k$ indicates which channel the patch belongs to and $i$ its raster position. After projecting the patches with a linear embedding layer and prepending the class tokens one obtains:

$$h = [x_{\text{CLS}} \quad h_{1,1} \quad \cdots \quad h_{1,N} \quad \cdots \quad h_{K,1} \quad \cdots \quad h_{K,N}] \in \mathbb{R}^{D \times (1+NK)},$$

As a result, the variable number of channels problem becomes a variable sequence length one, benefiting from the transformer's inherent capability of handling arbitrary sequence lengths.

**Channel embeddings.**    In the single-channel patch model, the approach of Bourriez et al. (2024) and Bao et al. (2024) to retain channel information, as well as position information, is to add both position embeddings and channel embeddings to the patch embeddings $h_{k,i}$. Let $p_i$ $(i = 1, \ldots, N)$ denote the position embeddings and $c_k$ $(k = 1, \ldots, K)$ the channel embeddings. The resulting sequence of patch embeddings for a sample $x$ is:

$$[x_{\text{CLS}} \quad h_{1,1} + p_1 + c_1 \quad \cdots \quad h_{K,N} + p_N + c_K].$$

Note that, if the maximum number of channels per image on the pre-training data is $K_{\max}$, this method cannot be used on images that have more channels than $K_{\max}$, as there will be no trained channel embeddings for the extra channels.

Bourriez et al. (2024) used a ViT-S architecture and the DINO SSL objective. For fairness in comparison with DINO BoC, we scale the model using a ViT-L architecture and update the pre-training recipe to the improved DINOv2. This model will be referred to as the *Channel-ViT* model.

Note that Channel-ViT outputs a single constant-sized embedding regardless of the channel-composition of the input data, contrary to the strategy of obtaining separate features for each channel. The utility of producing a constant-sized embedding irrespective of the number of channels is questionable (see Appendix G). We also tested a channel sampling technique introduced by Bao et al. (2024) for the training of Channel-ViT, however it did not improve the performance of the model, as shown in Appendix E.

Bourriez et al. (2024) claimed superiority of the joint-channel-encoding strategy over independent-encoding, attributing it to the inter-channel attention mechanism. However we have found that, at scale, combining independent-channel-encoding with the DINOv2 SSL method brings superior performance and robustness.

### 3.2.1 Hierarchical Attention Model

This work introduces a novel approach that balances joint and independent-channel-encoding strategies. This method plays a critical role in testing the hypothesis that limiting inter-channel interactions enhances the performance of channel-invariant models.

In this approach, the image is reshaped into a sequence of single-channel patches $x \in \mathbb{R}^{S^2 \times NK}$. After embedding the patch tokens with a linear layer, both a global class token $x_{\text{CLS}}$ and channel class tokens $x_{\text{CH}}$ are inserted into the sequence, only position embeddings are used:

$$[x_{\text{CLS}} \quad x_{\text{CH}_1} \quad h_{1,1} + p_1 \quad \cdots \quad h_{1,N} + p_N \quad \cdots \quad x_{\text{CH}_K} \quad h_{K,1} + p_1 \quad \cdots \quad h_{K,N} + p_N].$$

A specialized attention mask is employed in the Multi-Head Self-Attention blocks. Tokens within a single channel (channel class token and corresponding patch tokens) can only attend to other tokens within that same channel, therefore, at this level, the channels are processed independently. The global class token, however, attends to all channel class tokens, enabling inter-channel reasoning at a higher semantic level. This approach, termed DINO *Hierarchical Attention* (DINO HA) model, and the attention mask are illustrated in Figure 2 (see Appendix C for more details).

## 4 EXPERIMENTS

We test the merits of DINO BoC on diverse biological benchmarks, and compare it to existing channel-invariant strategies. Section 4.1 introduces the datasets used in this work, including the CHAMMI benchmark. Section 4.3 demonstrates the impact of choosing the SSL DINOv2 method (Oquab et al., 2023) instead of MAE (He et al., 2022), and compares DINO BoC to Channel-ViT and Microsnoop (Xun et al., 2024). In section 4.4, we further investigate the advantages of DINO BoC on cross-dataset generalization tasks. Then, in section 4.5 we evaluate self-supervised DINO BoC, Channel-ViT, and DINO HA models in the CHAMMI benchmark – designed to assess performance in in-distribution and OOD tasks – compared to SOTA supervised channel-invariant approaches.

### 4.1 DATASETS AND BENCHMARKS

We leverage multiple microscopy datasets with varying numbers of channels. In particular, we use the Human Protein Atlas, WTC-11, JUMP-CP and Cyclops datasets for evaluation tasks. Additionally, we employ the CHAMMI benchmark, a standardized evaluation framework for channel-invariant models.

**Human Protein Atlas dataset.** The subset of the Human Protein Atlas (HPA) data that is considered is the one of the Kaggle competition Human Protein Atlas (2019), concerned with the subcellular distribution of the proteins encoded by different genes. It covers 35 cell lines and 28 subcellular structures of protein localization. There are $113, 545$ images in total, with four channels. There is also a single cell version of the same dataset (Le et al., 2022), obtained through segmentation of the field-of-view (FOV) images. The *HPA Single Cell* dataset contains $839, 612$ images.

**WTC-11 dataset.** This dataset is a version of the WTC-11 hiPSC Single-Cell Image Dataset v1 (Viana et al., 2023) of the Allen Institute curated for the CytoData Symposium 2022 hackathon (Allen Institute, 2022). The dataset contains $214, 037$ 3D images of cells, we used the maximum z-projection of the original images. The dataset provides cell-cycle stage annotations, with six stages. The images have one bright-field (BF) channel and three fluorescence channels; BF was excluded.

**Cell Painting dataset.** We utilize the Cell Painting (CP) dataset curated by Moshkov et al. (2024), totaling $8, 423, 455$ images with five channels. The dataset has the objective of allowing the study of the response of cells to different compound treatments and gene over-expression experiments.

**JUMP-CP dataset.** This dataset, used by Bao et al. (2024), is a processed version of the data made available by the JUMP-Cell Painting Consortium (Broad Institute, 2021). It contains $229, 228$ single cell images. We used only the five fluorescence channels in our work.

**Cyclops dataset.** We used the same dataset as described by Xun et al. (2024) consisting of 28,166 2-channel yeast cell images from the Cyclops database (Lu et al., 2018).

**OpenCell dataset.** The OpenCell dataset was introduced by Kobayashi et al. (2022), and encompasses $1, 311$ different tagged proteins. In total, $1, 134, 592$ images were made available, with two fluorescence channels. More details on the five datasets mentioned above are given in Appendix A.

**CHAMMI benchmark.** The CHAMMI benchmark (Chen et al., 2024) includes a dataset curated from the WTC-11, HPA Single Cell and Cell Painting datasets. In total there are $220, 284$ images, of which $100, 145$ are used for training. The benchmark is a standardized evaluation framework for channel-invariant models. It presents a comprehensive set of nine tasks for channel-invariant models of varying complexity, that evaluate the ability of the models to generalize to new biologically-relevant experimental regimes. As such it positions itself as an indispensable benchmark to evaluate those models. The images from each data source present in the CHAMMI dataset are split into one training set and several test sets, designed for specific tasks. Tasks with suffix 1 are IID classification problems, where the test and train data follow the same distribution. Tasks with suffix greater than 1 evaluate

the OOD generalization capabilities of the model, and simulate biologically-relevant application scenarios (see Appendix B.1 for a detailed description of each task).

**ExtendedCHAMMI dataset** We extend the CHAMMI train set to a total of $7,748,662$ images, incorporating additional data from both the source datasets and new data sources. The extended training dataset preserves the OOD characteristics of the CHAMMI tasks (see Appendix B.2).

## 4.2 IMPLEMENTATION DETAILS

When pre-training the models, care was taken to ensure that the models processed the same amount of data. For the DINO BoC model, a sample consists of a single channel, whereas for the other models, a sample is an image with all of its channels. Therefore the former must be trained for more iterations to achieve fair comparison.

For each pre-training dataset, the Channel-ViT and DINO HA models were pre-trained for $45,000$ iterations. Taking into account the average number of channels in the pre-training dataset, the Bag of Channels model was trained for a proportionally larger number of epochs. The batch size used was of $1024$ for all models, and the batch size per GPU was set to 8, except for DINO BoC model, for which 32 fits in memory. On our largest dataset, we trained the models for about 2 days using 16 nodes, or 4 nodes for the DINO BoC model. Unless specified otherwise, we trained ViT large models. More details are provided in Appendix D. On the ExtendedCHAMMI dataset, the channel-invariant models were pre-trained with balanced sampling across data sources.

## 4.3 SCALING CHANNEL AGNOSTIC FEATURE REPRESENTATIONS WITH DINOV2

Table 1 displays the results for channel-invariant models pre-trained on the ExtendedCHAMMI dataset, as well as for baseline fixed-channel models (Doron et al., 2023) pre-trained either on the HPA-FOV, JUMP-CP or WTC dataset. We evaluate the models on the HPA-FOV, JUMP-CP and WTC datasets, note that the fixed-channel models can only be evaluated on the datasets they are pre-trained on. The JUMP-CP dataset is not included in ExtendedCHAMMI, therefore it evaluates the generalization capability of channel-invariant models on novel data sources. Results for an ablation removing datasets from the ExtendedCHAMMI dataset are presented in Appendix H, and results on all eight JUMP-CP channels are listed in Appendix I.

First of all – comparing DINO BoC to Channel-VIT using the same SSL method and network size – we observe that the strategy of independently encoding the channels significantly outperforms the one of jointly encoding them across all tasks.

We also observe that DINO BoC has stronger performance than fixed-channel models on three out of four tasks, including when evaluating on the novel JUMP-CP dataset. This shows that DINO BoC successfully leverages diverse microscopy data to learn an improved encoder, justifying the interest in channel-invariant models.

Table 1 also shows that DINO outperforms MAE as a learning objective, demonstrating that DINOv2 is a key component of the success of our approach.

Furthermore, we evaluate the impact of the network size on DINO BoC. Although using a ViT-L leads to improved performance, even with a ViT-S, DINO BoC outperforms ViT-L MAE BoC and Channel-ViT models.

Absent the ability of directly pre-training Microsnoop on ExtendedCHAMMI due to the unavailability of the training code, the performance of MAE BoC serves as a proxy for Microsnoop performance, as it uses the same SSL method. In addition, we demonstrate in Table 2 that DINO BoC outperforms Microsnoop on the challenging Cyclops dataset (on which they reported the largest gains) by a substantial margin, while especially fortifying performance on rare classes (up to 20 points).

Unexpectedly, the BoC approach, and DINO BoC in particular, thus not only matches, but outperforms joint-channel-encoding, including on protein-localization prediction, setting a new SOTA.

Table 1: **Comparison of channel-invariant models trained on the ExtendedCHAMMI dataset and fixed-channel models.** The first three rows (CellDINO) are fixed-channel baseline models, separately pre-trained either on the HPA-FOV, JUMP-CP or WTC datasets. Best channel-invariant results in bold; best results overall are underlined.

| Model | SSL method | Network size | Channel invariant | Training set | HPA-FOV F1 Protein loc. | HPA-FOV F1 Cell type | JUMP-CP Accuracy | WTC F1 Cell cycle st. |
|---|---|---|---|---|---|---|---|---|
| CellDINOv2 | DINOv2 | ViT-L | ✗ | HPA-FOV | 65.0 | 89.3 | ✗ | ✗ |
| CellDINOv2 | DINOv2 | ViT-L | ✗ | JUMP-CP | ✗ | ✗ | 44.3 | ✗ |
| CellDINOv1 | DINOv1 | ViT-L | ✗ | WTC | ✗ | ✗ | ✗ | 82.3 |
| Channel-ViT | DINOv2 | ViT-L | ✓ | ExtendedCHAMMI | 57.4 -7.6 | 90.4 +1.1 | 39.4 -4.9 | 87.2 +4.9 |
| BoC | MAE | ViT-L | ✓ | ExtendedCHAMMI | 54.0 -11.0 | 90.8 +1.5 | 39.3 -5.0 | 89.4 +7.1 |
| | DINOv2 | ViT-S | ✓ | | 55.6 -9.4 | 90.7 +1.4 | 44.5 +0.2 | **91.0** +8.7 |
| | DINOv2 | ViT-L | ✓ | | **61.7** -3.3 | **91.1** +1.8 | **45.2** +0.9 | 90.5 +8.2 |

Table 2: **Comparison to Microsnoop on the Cyclops dataset.** DINO BoC dramatically outperforms Microsnoop, especially on the 4 least frequent classes (out of 16).

| Class frequency | Budtip 1.5 % | Cell periphery 1.9% | Budneck 2.4% | Actin 3.8% | All |
|---|---|---|---|---|---|
| Microsnoop (Xun et al., 2024) | 32.1 | 96.4 | 43.4 | 48.0 | 75.9 |
| DINO BoC | **62.1** | **97.5** | **72.6** | **63.7** | **83.1** |

## 4.4 CROSS-DATASET GENERALIZATION

To further analyze the surprising result that inter-channel reasoning is detrimental to the performance and robustness of channel-invariant models we investigate the cross-dataset generalization capabilities of the DINO Channel-VIT, DINO HA and DINO BoC models.

We train the models either on HPA-FOV or JUMP-CP, and evaluate them on HPA-FOV, JUMP-CP and WTC, the results are shown in Table 3. DINO BoC outperforms DINO Channel-ViT on all cross-dataset tasks. A further point of analysis is the DINO HA model, which balances joint and independent-channel-encoding characteristics. DINO HA yields systematically better performances than Channel-ViT, while falling behind DINO BoC, corroborating the conclusion that independent-channel-encoding is the winning strategy for channel-invariant models.

Table 3: **Cross-dataset generalization of channel-invariant models**. DINO BoC shows superior performance on unseen channel combinations. In-dataset results are shown in gray for reference.

| Model | Channel invariant | Training set | HPA-FOV F1 Protein loc. | HPA-FOV F1 Cell type | JUMP-CP Accuracy | WTC F1 Cell cycle st. |
|---|---|---|---|---|---|---|
| DINO Channel-ViT | ✓ | HPA-FOV | 65.5 | 90.9 | 35.3 | 80.0 |
| DINO HA (Ours) | ✓ | HPA-FOV | 66.7 | 91.3 | 37.3 | 88.9 |
| DINO BoC (Ours) | ✓ | HPA-FOV | 65.2 | 91.5 | **40.2** | **89.8** |
| DINO Channel-ViT | ✓ | JUMP-CP | 29.5 | 82.0 | 53.4 | 81.8 |
| DINO HA (Ours) | ✓ | JUMP-CP | 30.3 | **85.2** | 52.0 | 84.2 |
| DINO BoC (Ours) | ✓ | JUMP-CP | **31.6** | 85.0 | 41.3 | **90.5** |

## 4.5 OUT-OF-DISTRIBUTION GENERALIZATION ON CHAMMI

Table 4 reports results on the CHAMMI benchmark using the 1-NN classifier protocol as defined in (Chen et al., 2024). The table provides a comparison to the results of Chen et al. (2024) for supervised models trained from scratch, where the best performance is obtained by the HyperNet

model. The approach, inspired by Hypernetworks (Ha et al., 2017), uses a CNN backbone and a MLP that generates kernel weights for the initial convolutional layer of each input channel. Note that while the model is channel-invariant, it is supervised and requires labels during training, as is the case for all other channel-invariant models in CHAMMI (Chen et al., 2024). Despite the small size of the CHAMMI training set of about 100k images, not ideal for pre-training, Table 4 demonstrates improved F1 scores of our DINO BoC approach on OOD tasks on two datasets out of three, outperforming the best supervised baseline by $5.3\%$ on WTC and $3,8\%$ on CP on average.

### 4.5.1 SCALING SELF-SUPERVISED CHANNEL-INVARIANT APPROACHES

SSL methods tend to benefit from pre-training on a larger corpus of data. To explore the scaling properties for channel-invariant models, we hence leverage the ExtendedCHAMMI dataset, which has almost 8M images and preserves the OOD characteristics of the CHAMMI tasks.

The flexibility to extend the training set with new images with no labels is an advantage of SSL pre-training. In contrast, supervised methods are constrained by the need for more annotated data. Table 5 presents results for the SSL models pre-trained on the ExtendedCHAMMI dataset.

Table 4: **F1 scores for 1-NN search on the CHAMMI test set. The models were pre-trained on the CHAMMI train split.** Lines 1-6 report the results of Chen et al. (2024) for CNN-based models trained from scratch in a *supervised* fashion. Line 1 reports the performance of FixedChannels, that consists on a separate model trained for each fixed channel combination. Lines 2-6 are channel-invariant models. Lines 7-9 are self-supervised channel-invariant ViTs. Best results between channel-invariant self-supervised approaches are in bold.

| Model | Mean | Average OOD WTC | HPA | CP | WTC Task1 | Task2 | HPA Task1 | Task2 | Task3 | CP Task1 | Task2 | Task3 | Task4 |
|---|---|---|---|---|---|---|---|---|---|---|---|---|---|
| FixedChannels | 50.0 | 64.8 | 59.2 | 25.9 | 64.9 | 64.8 | 80.7 | 76.3 | 42.1 | 66.0 | 48.1 | 23.0 | 6.6 |
| Depthwise | 51.7 | 65.2 | 64.4 | 25.6 | 68.9 | 65.2 | 84.9 | 81.3 | 47.5 | 67.3 | 47.8 | 22.4 | 6.5 |
| TargetParam | 49.6 | 59.0 | 62.3 | 27.3 | 69.5 | 59.0 | 83.7 | 79.4 | 45.2 | 71.7 | 50.8 | 23.4 | 7.7 |
| SliceParam | 45.7 | 56.8 | 54.6 | 25.6 | 61.6 | 56.8 | 77.0 | 69.0 | 40.3 | 64.6 | 47.5 | 22.2 | 7.1 |
| HyperNet | 53.7 | 66.1 | 67.1 | 27.8 | 72.6 | 66.1 | 88.7 | 85.8 | 48.3 | 72.0 | 51.7 | 24.7 | 6.9 |
| Template mixing | 46.6 | 56.5 | 57.7 | 25.7 | 63.1 | 56.5 | 80.8 | 74.1 | 41.3 | 67.1 | 46.8 | 22.7 | 7.5 |
| DINO Channel-ViT | 42.6 | 45.3 | **53.6** | 29.0 | 68.7 | 45.3 | **92.2** | **65.2** | **42.0** | **95.2** | 51.5 | **25.2** | 10.3 |
| DINO BoC (Ours) | **48.8** | **71.4** | 43.3 | **31.6** | **79.4** | **71.4** | 87.0 | 56.4 | 30.2 | 93.5 | **58.5** | 20.1 | **16.3** |

Table 5: **F1 scores for 1-NN search on the CHAMMI test set. The self-supervised models were pre-trained on the ExtendedCHAMMI dataset.** Line 1 presents the best performing *supervised* baseline (HyperNet), which can only be trained on the annotated subset of CHAMMI.

| Model | Mean | Average OOD WTC | HPA | CP | WTC Task 1 | Task 2 | HPA Task 1 | Task 2 | Task 3 | CP Task 1 | Task 2 | Task 3 | Task 4 |
|---|---|---|---|---|---|---|---|---|---|---|---|---|---|
| HyperNet | 53.7 | 66.1 | 67.1 | 27.8 | 72.6 | 66.1 | 88.7 | 85.8 | 48.3 | 72.0 | 51.7 | 24.7 | 6.9 |
| DINO Channel-ViT | 43.6 | 46.2 | **55.6** | 28.9 | 64.5 | 46.2 | **92.1** | **65.3** | **45.9** | 89.0 | 53.5 | 21.8 | 11.3 |
| DINO BoC (Ours) | **51.6** | **79.0** | 43.0 | **32.7** | **79.4** | **79.0** | 86.6 | 59.3 | 29.6 | **92.6** | **57.6** | 22.1 | **18.5** |

Comparing Tables 4 and 5, it is evident that the models benefit from scaling the dataset size, even though part of the additional data comes from datasets unrelated to those on which the models are evaluated on. DINO BoC show the greatest improvements, gaining 2.8 points in the average OOD score, while Channel-ViT gains only 1.0 point. This is a promising result for channel-invariant models, especially for DINO BoC, demonstrating that by leveraging diverse data sources, more robust biological feature extractors can be learned, bringing advantages over study-specific models.

### 4.5.2 LINEAR PROBE FOR CHAMMI

Analyzing Table 5, among the self-supervised models, DINO BoC has the best performance. Furthermore, the OOD results of DINO BoC strongly outperform all previous attempts on the WTC and CP datasets, including the supervised HyperNet strategy. This result suggests that self-supervision can help overcome the limitations of supervised learning for OOD generalization. Supervised learning

may capture spurious correlations in the training set, leading to poor performance on OOD tasks. In contrast, SSL leverages only image-based information, resulting in an unbiased representation that can be more robust to domain shifts.

We note that the Channel-ViT and DINO BoC models were pre-trained in a self-supervised fashion, while the results reported in the CHAMMI paper (Chen et al., 2024) reflect models pre-trained with a supervised ProxyNCA++ loss (Teh et al., 2020), which represents each training class with a proxy in the embedding space, and draws the samples towards their corresponding proxies. It therefore naturally encourages the clustering of the data in the embedding space according to their labels, facilitating the one nearest neighbor search. Without the benefit of label information to organize the embedding space according to a given downstream task, SSL approaches yield nested embedding spaces (Doron et al., 2023). For example, for the HPA task, the features first cluster by cell type, while the protein localization are retained as a nested factor of variation (see Appendix F). This organization is ill-suited to nearest-neighbor protein localization classification on a novel cell type (HPA Task 2), or on a known cell type but for which there are no examples of the targeted protein localization (HPA Task 3). We hence further evaluated model performance using a linear probe.

Table 6: **F1 scores for a linear probe on CHAMMI test set.** Self-supervised models were pre-trained on the ExtendedCHAMMI dataset, while the supervised HyperNet was pre-trained on CHAMMI.

| Model | Average OOD | | | | WTC | | HPA | | | CP | | | |
|---|---|---|---|---|---|---|---|---|---|---|---|---|---|
| | Mean | WTC | HPA | CP | Task1 | Task2 | Task1 | Task2 | Task 3 | Task1 | Task2 | Task3 | Task4 |
| HyperNet | 65.4 | 85.3 | 82.9 | 27.9 | 87.9 | 85.3 | 94.4 | 92.5 | 73.2 | 93.5 | 51.5 | 17.3 | 15.0 |
| DINO Channel-ViT | 59.8 | 66.9 | **76.7** | 35.9 | 83.1 | 66.9 | 88.2 | **84.9** | 68.4 | 80.5 | 54.5 | 23.3 | 30.0 |
| DINO HA (Ours) | 62.7 | 76.2 | 72.4 | 39.5 | 88.0 | 76.2 | **88.5** | 82.4 | 62.4 | **91.7** | **61.6** | **27.5** | 29.3 |
| DINO BoC (Ours) | **67.9** | **89.2** | 74.9 | **39.7** | **90.5** | **89.2** | 88.3 | 84.7 | 65.0 | 90.5 | 60.5 | 25.8 | **32.7** |

Moreover, the use of a linear probe is of particular interest for DINO BoC: while no explicit cross-channel features can be learned by models that encode channels independently, relevant information may nevertheless be preserved. Thus, even a minimal opportunity to relate information across channels may lead to further performance gains. Indeed, comparing Tables 5 and 6, the use of a linear probe significantly improves the scores on HPA Task 2 and 3, corroborating the hypothesis that the poor performance of nearest neighbor search on these tasks is due to the nesting of the factors of variation. Moreover, employing a linear classifier, DINO BoC surpasses the HyperNet supervised baseline on the mean OOD score, as well as Channel-ViT on HPA, suggesting that ample information suitable to cross-channel integration is preserved in its channel-specific embeddings. This highlights the potential of this simple strategy to yield a powerful biological feature extractor.

## 5 CONCLUSIONS

We report results on a large-scale study into self-supervised channel-invariant training strategies, as a step towards general-purpose feature extractors for fluorescent microscopy. Scaling the BoC approach using Vision Transformers and the state-of-the-art self-supervised DINOv2 method, DINO BoC outperforms models that rely on inter-channel reasoning, and positions it as the leading channel-invariant approach. In addition to its strong performances, DINO BoC is notable for its simplicity, lacking any priors; whereas joint-encoding methods can *adapt* to variable channel numbers up to some maximum (see Appendix I), DINO BoC is channel-agnostic, rendering it suitable to arbitrary channel combinations. We also note that even the theoretical advantage of a uniform embedding space produced by joint-encoding methods (Bourriez et al., 2024) compared to BoC in practice remains unclear (see Appendix G). Instead, we show that DINO BoC substantially outperforms Channel-ViT in generalization to unseen channel combinations, and OOD tasks at test time. More broadly, our results call the utility of joint-channel-encoding as a prior for non-RGB domains into question. Indeed, we find that the DINO BoC approach achieves performance on par with SOTA results out-of-the-box for aerial remote sensing settings (Appendix J).

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

## A  DATASETS

This section provides a detailed description of the datasets, and of the channels they encompass.

**Human Protein Atlas dataset.**  The Human Protein Atlas (HPA) is an initiative that aims to map all human proteins across cells, tissues and organs. The subset of the data that is considered is the one of the Kaggle competition Human Protein Atlas (2019), concerned with the subcellular distribution of the proteins encoded by different genes. It covers 35 cell lines and 28 subcellular structures of protein localization. It covers thirty five cell lines and more than $13,000$ proteins. The subcellular localization of each protein was classified into one or more of 28 subcellular structures.

The images were acquired using immunofluorescence and confocal microscopy. Four fluorescence dyes binding to (0) microtubules, (1) encoded protein, (3) nucleus and (4) endoplasmic reticulum are imaged in different channels. There are $113,545$ images in total.

There is also a single cell version of the same dataset Le et al. (2022) obtained through segmentation of the FOV images. The *HPA Single Cell* dataset contains $839,612$ images.

**WTC-11 dataset.**  The dataset contains $214,037$ 3D images of cells, spanning 25 cellular structures. Other than tagging the structure of interest with a fluorescent protein (FP), fluorescent DNA and cell-membrane dyes were employed. The images have four-channels: bright-field; nucleus; cell membrane; structure of interest. Given that the focus of this work is to develop a foundation channel-invariant model for fluorescent microscopy, the bright-field channel was discarded.

The dataset provides cell-cycle stage annotations. The six possible labels are M0, M1M2, M3, M4M5, M6M7_single, M6M7_complete; where M0 through M7 denote cell cycle stages.

**Cell Painting dataset.**  The Cell Painting Dataset Doron et al. (2023) used is the combination of the LINCS (Way et al., 2022), BBBC036 (Bray et al., 2017) and a third curated dataset (Moshkov et al., 2024), which includes BBBC022. All of those datasets were obtained following the Cell Painting protocol Bray et al. (2016), a standardized morphological profiling assay that images six fluorescent dyes in five channels, revealing eight cellular components. The components visualized in each channel are (0) nucleus; (1) endoplasmic reticulum; (2) nucleoli, cytoplasmic reticulum; (3) actin, golgi, plasma membrane; and (4) mitochondria. The goal of the studies included in the Cell Painting Dataset was to quantify the response of the cells to different perturbations: either compound treatments or gene over-expression experiments. Overall, the dataset includes $400$ compounds and $80$ gene over-expression experiments, totaling $8,423,455$ images.

**CHAMMI dataset.**  The CHAMMI dataset was curated from the WTC-11, HPA Single Cell and Cell Painting datasets. It includes $65,103$ images from the WTC-11 dataset covering six tagged structures; $66,936$ images from the HPA Single-Cell dataset covering $18$ cell lines and $8$ protein localization classes, only images with a single protein localization annotation were selected; and $88,245$ images from the Cell Painting dataset covering seven compound experiments, including the negative control. In total there are $220,284$ images, of which $100,145$ are used for training.

**JUMP-CP dataset.**  We considered the version of the JUMP-CP dataset used by Bao et al. (2024); it is a processed version of the data made available by the JUMP-Cell Painting Consortium Broad Institute (2021). Each image includes the five Cell Painting channels and three brightfield channels (HighZBF, LowZBF and brightfield).

The datasets generated by the JUMP-Cell Painting Consortium have the goal of enabling image-based drug mechanisms of action determination. As such, it encompasses multiple chemical and genetic perturbations. This particular version of the JUMP-CP dataset contains $229,228$ single cell images.

**OpenCell dataset.**  The OpenCell dataset was introduced by Kobayashi et al. (2022), and consists of confocal images encompassing $1,311$ different tagged proteins. On average, each protein was imaged in 18.59 field of view images. Crops containing from 1 to 3 complete cells were extracted from each image, resulting in approximately $800$ cropped images per protein. In total, $1,134,592$ images were made available. In addition to the tagged protein, a nuclear marker was used to visualize

the nucleus. From the nuclear channel, they constructed a distance map and segmentation masks. However those two additional channels were not used for the purposes of this work.

# B    DETAILS ON THE CHAMMI BENCHMARK AND DATASET EXTENSION

## B.1    CHAMMI BENCHMARK

The CHAMMI benchmark Chen et al. (2024) is a standardized evaluation framework for channel-invariant models. It presents a comprehensive set of nine tasks for channel-invariant models of varying complexity, that evaluates the ability of the models to generalize to new biologically-relevant experimental regimes. The images from each data source present in the CHAMMI dataset are split into one training set and several test sets, designed for specific tasks. Tasks with suffix 1 are IID classification problems, where the test and train data follow the same distribution. Originally, the CHAMMI benchmark considers a Nearest Neighbor (NN) evaluation.

The WTC-11 data is used for cell-cycle stage classification. The train set contains images with one of four cellular structures tagged: nuclear speckles, mitochondria, microtubules, or Golgi apparatus. Images of *W_Task2* are tagged with three novel cellular structures, the task evaluates whether the model is able to classify cell-cycle stages when an unseen cellular structure is tagged.

The HPA data supports protein localization classification. The train split covers 17 cell lines and four protein localizations: nuclear speckles, mitochondria, microtubules, or Golgi apparatus. The *H_Task2* images come from a novel cell line but covering the same protein localizations as the train split. The *H_Task3* images come from the same cell lines as the train split, but labeled with one of three novel protein localizations.

Lastly, the Cell Painting data is used for perturbation classification. The train set includes images of cells coming from 9 plates and perturbed with one of three treatments, as well as negative controls; The *C_Task2* images are perturbed with the same treatments as the train split and coming from the same data sources, however they belong to a set of 3 novel plates. The *C_Task3* includes the same treatments as the train split, but coming from the BBBC022 dataset and covering 4 novel plates. Finally, the *C_Task4* images are from the same set of plates and data sources as the train split, but the cells are perturbed with novel treatments.

For tasks that introduce new labels that are not present in the train set (HPA Task 3 and CP Task 4), a leave-one-out evaluation strategy is employed. Taking the example of HPA Task 3, the test data is split into sub-groups according to the cell line. Then, for each sub-group, the NN search is computed on both the training data and the remaining sub-groups. For CP Task 4, the data is split by plate ID.

## B.2    EXTENDEDCHAMMI DATASET

The ExtendedCHAMMI dataset extends the CHAMMI train split to a total of $7,748,662$ images, using additional data from both the source datasets and new data sources, while preserving the OOD characteristics of the CHAMMI tasks.

In order to build the *ExtendedCHAMMI* dataset, the HPA FOV, HPA Single Cell, WTC-11, Cell Painting and OpenCell datasets were used. The samples belonging to the IID tasks W_Task1, H_Task1 and C_Task1 were removed from the WTC-11, HPA Single Cell and Cell Painting datasets, respectively. Furthermore, the images of the HPA FOV dataset containing cells present on H_Task1 were removed as well. With respect to the OOD tasks, the unseen tagged cellular structures for WTC-11; cell lines and protein localizations for HPA FOV and HPA Single Cell; and plates, data sources and treatments for Cell Painting were removed. The resulting number of images per dataset is summarized in Table 7.

# C    HIERARCHICAL ATTENTION MODEL TRAINING OBJECTIVE

The hierarchical attention model is based on a single-channel patch approach and, in addition to a global CLS token, it also inserts into the sequence channel CLS tokens. Moreover, it leverages a hierarchical attention mask (Figure 2), that constrains a channel's patch and CLS tokens to only

Table 7: **Data included on the ExtendedCHAMMI dataset.** The third column lists the amount of data that was included on ExtendedCHAMMI over the total size of the data source. For each data source, the number of channels and image type is specified: field-of-view (FOV) images, cropped FOV images containing a smaller number of cells, or images of a single cell. The last column lists which biological or experimental factors were discarded from the original datasets to preserve the OOD characteristic of the CHAMMI tasks.

| Dataset | Image type | Size | Channels | Discarded factors |
|---|---|---|---|---|
| **HPA Single Cell** | One cell | 296670/839612 | 4 | Cell line (HEK 293); protein localization (cytosol, endoplasmic reticulum, nucleoplasm) |
| **WTC** | One cell | 179994/214037 | 3 | Tagged cellular components (centrioles, tight junctions, actin bundles) |
| **OpenCell** | Cropped FOV | 1134592/1134592 | 2 | None |
| **Cell Painting** | Cropped FOV | 6103565/8423455 | 5 | Plates (SQ00015125, SQ00015168, SQ00015221); data source (BBBC022); treatments (BRD-K11129031, BRD-K62310379, BRD-K77947974) |
| **HPA FOV** | FOV | 33841/102190 | 4 | Cell line (HEK 293); protein localization (cytosol, endoplasmic reticulum, nucleoplasm) |

interact with one another, while the global CLS token attends to the channel CLS tokens. Consequently, the model produces both a global image representation, and channel-level representations. In view of this, additional loss terms were included in the DINOv2 training objective, to account for the channel-level representations.

Consider an image $x$, and let $\mathcal{G}(x)$ denote the set of global crops of $x$, and $\mathcal{C}(x)$ the set of all crops of $x$, note that $\mathcal{G}(x) \subset \mathcal{C}(x)$. Furthermore, let $p_s^{[\text{CLS}]}(u)$ and $p_t^{[\text{CLS}]}(u)$ be the CLS tokens, transformed into probability vectors, output by the student and teacher networks for a crop $u$. Then, the DINO loss for a sample $x$ is:

$$\sum_{\substack{u \in \mathcal{G}(x)}} \sum_{\substack{v \in \mathcal{C}(x) \\ v \neq u}} H\left(p_t^{[\text{CLS}]}(u), p_s^{[\text{CLS}]}(v)\right),$$

where $H(\cdot)$ denotes the cross-entropy. With the additional channel CLS tokens, $[\text{CH}_1], \ldots, [\text{CH}_C]$, the DINO loss is extended to:

$$\sum_{\substack{u \in \mathcal{G}(x)}} \sum_{\substack{v \in \mathcal{C}(x) \\ v \neq u}} \left(\lambda_{\text{dino}}^{[\text{CLS}]} H\left(p_t^{[\text{CLS}]}(u), p_s^{[\text{CLS}]}(v)\right) + \lambda_{\text{dino}}^{[\text{CH}]} \sum_{i=1}^{C} H\left(p_t^{[\text{CH}_i]}(u), p_s^{[\text{CH}_i]}(v)\right)\right).$$

Another component of the DINOv2 loss is the KoLeo regularizer, that encourages the uniform span of features within a batch. Other than applying the KoLeo loss to the set of global CLS tokens for a batch, the loss is also separately applied to the set of all channel CLS tokens in the batch, with weights $\lambda_{\text{koleo}}^{[\text{CLS}]}$ and $\lambda_{\text{koleo}}^{[\text{CH}]}$. The remaining component of the DINOv2 loss, the iBOT masked-image-modeling loss, is left unchanged. We gave an equal weight to the losses on the global and channel CLS tokens. For the pre-training on the small scale CHAMMI dataset only, the DINO and KoLeo losses on channel class tokens were discarded, due to instabilities during training.

# D  TRAINING AND EVALUATION DETAILS

## D.1  PRE-TRAINING DETAILS

Unless specified otherwise, we trained ViT-Large models with default patch size 16, and the default parameters of DINOv2 except a drop path rate of 0.1, a teacher momentum of 0.996, a learning rate of $5.0 \cdot 10^{-4}$, and 20 warm-up epochs. Transforms used include random contrast and brightness augmentations, flips and random resize crops of size 224.

## D.2 EVALUATION DETAILS

The feature vector used in the evaluations – with the exception of the evaluations on the CHAMMI benchmark – are obtained by concatenating the CLS tokens across the last $L$ layers ($L = 4$), as well as the channel-wise average pooled patch tokens. If $D$ is the dimension of the tokens (for a ViT-L $D = 1024$), and $K$ is the number of channels, then the feature size for Channel-ViT is $LD + KD$, for DINO BoC it is $LKD + KD$ and for DINO HA it is $L(1 + K)D + KD$.

For the evaluations on the CHAMMI benchmark (Tables 4, 5 and 6), only the CLS tokens are used and $L = 1$, therefore the feature size for Channel-ViT is $D$, for DINO BoC it is $KD$ and for DINO HA it is $(1 + K)D$. We ablate the impact of feature dimension in Appendix K.

We used the AdamW optimizer and a one cycle Cosine scheduler. We used the same train/val/test splits as Bao et al. (2024) for the JUMP-CP dataset. For HPA, we used the same train/val splits as Doron et al. (2023). For WTC, we created $80\% - 10\% - 10\%$ uniformly distributed train/val/test splits. For every evaluation, we trained 14 classifiers varying the learning rate between $10^{-4}$ and 1, and selected the best classifier on the val set. We trained all classifiers for 4350 iterations on 8 GPUs with a batch size per gpu of 32 for HPA-FOV and WTC, and 128 for JUMP-CP. To train the linear classifiers on HPA-FOV, the following transforms are used : random crop of size 384, flips and self normalization. For evaluation, a center crop of size $384 \times 384$ is taken, followed by self normalization. For JUMP-CP, we used the same normalization as in Bao et al. (2024) instead of self normalization, and crops of size 224. For WTC, we used self normalization and crops of size 224.

## E   INFLUENCE OF HIERARCHICAL CHANNEL SAMPLING

Bao et al. (2024) introduced a channel sampling technique for the training of ChannelViT, denoted hierarchical channel sampling (HCS). For an image $x$ with $K$ channels, HCS consists in sampling a number $m \in \{1, \ldots, K\}$ uniformly at random and then randomly selecting $m$ channels from $x$ without replacement.

Results summarized in Table 8 suggest that Hierarchical Channel Sampling (Bao et al., 2024) hinders the performance of single-channel patch models. The channel sampling technique was found by Bao et al. (2024) to boost performance when pre-training and evaluating on the same dataset, mainly in the supervised scenario of missing channels at evaluation time. However it does not translate into improvements on the channel heterogeneous setting explored in this work. We postulate that it plays the role of a regularizer in the supervised context, but that this strategy is not well adapted to SSL.

Table 8: **Influence of HSC on DINO Channel-ViT models.** The models were pre-trained on the ExtendedCHAMMI dataset and evaluated on CHAMMI.

| Model | HCS | Average OOD | | | | WTC | | HPA | | | CP | | | |
|---|---|---|---|---|---|---|---|---|---|---|---|---|---|---|
| | | Mean | WTC | HPA | CP | Task1 | Task2 | Task1 | Task2 | Task3 | Task1 | Task2 | Task3 | Task4 |
| DINO Channel-ViT | ✗ | **43.6** | **46.2** | **55.6** | **28.9** | 64.5 | **46.2** | **92.1** | **65.3** | **45.9** | 89.0 | 53.5 | 21.8 | **11.3** |
| DINO Channel-ViT | ✓ | 39.9 | 39.5 | 51.9 | 28.4 | **66.4** | 39.5 | 88.5 | 62.3 | 41.5 | **90.2** | **56.0** | **22.3** | 6.9 |

## F   HIERARCHY OF FACTORS OF VARIATION IN THE FEATURE SPACE

SSL methods learn image representations using the samples themselves as the supervisory signal, therefore no label information controls the organization of the features in the embedding space.

In the channel-invariant models pre-trained on HPA FOV, we observe an emerging clustering of the features according to a hierarchy of semantic concepts, as illustrated in Figure 3. The features first cluster by cell type, while the protein localization is retained as a nested factor of variation.

## G   CHANNELS AS CONFOUNDERS IN A UNIFIED FEATURE SPACE

To assess the utility of a unified feature space produced by channel-invariant models such as the one proposed by Bourriez et al. (2024), we explore the effect of ablating channels in the HPA FOV

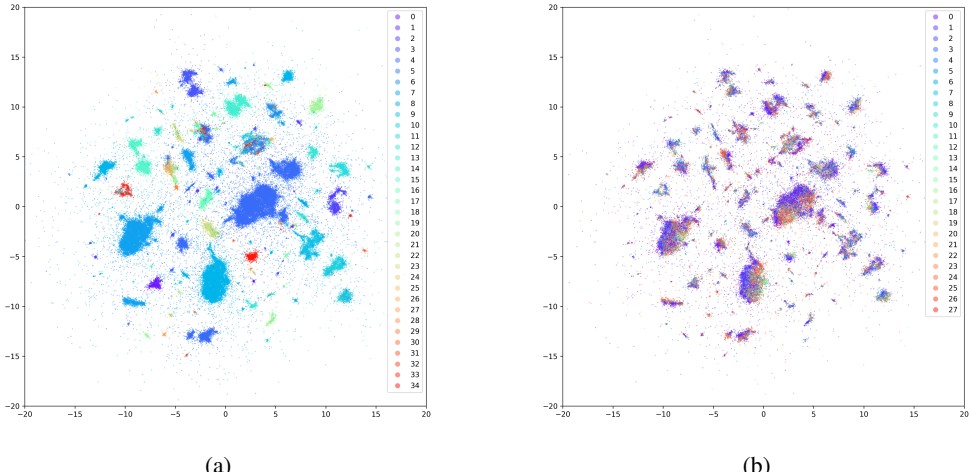

(a)                                                          (b)

Figure 3: **UMAP of the HPA FOV dataset highlighting different factors of variation.** UMAP space of the HPA FOV features obtained from the DINO BoC model pre-trained on the same dataset, colored according to (a) cell type and (b) protein localization. In (b) only samples with a single protein localization are displayed. The features are obtained separately for each channel, and then concatenated.

dataset. Specifically, we remove nucleus and ER channels from a random half of the dataset and compare the resulting features against those of images with all channels within a jointly computed UMAP space (Figure 4).

We observe that the data is clustered into distinct groups depending on the channels, and samples with different channels can hardly be compared to one another, even if their features have the same dimension. Therefore a common embedding space does not constitute an advantage of Channel-ViT over DINO BoC.

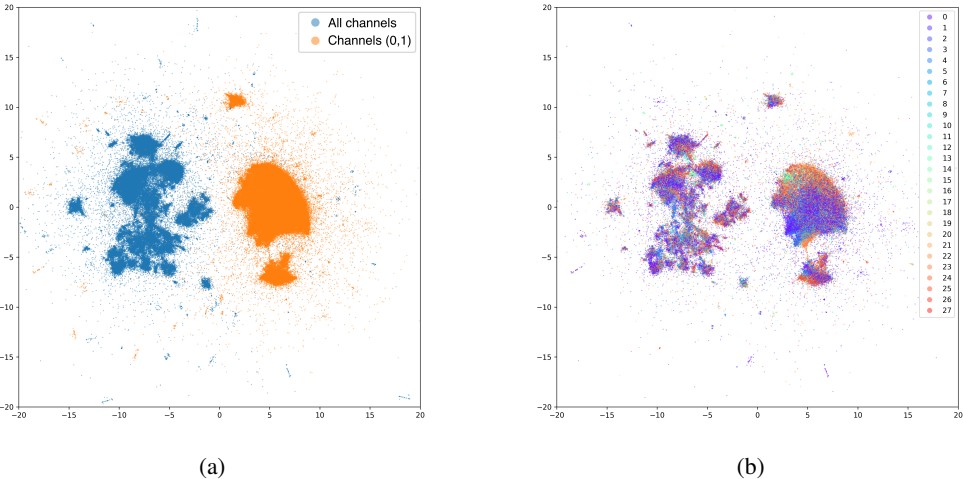

(a)                                                          (b)

Figure 4: **UMAP of the HPA FOV dataset to assess utility of unified feature space.** a) UMAP space of the HPA FOV features obtained from the Channel-ViT model pre-trained on the same dataset, comparing features computed when the model sees all four channels of the dataset, in blue, and features computed with only two channels (microtobules and protein), in orange. b) Same UMAP as in (a) but filtered for images with only one protein localization and colored according to them.

## H   IMPACT OF THE REMOVAL OF DIFFERENT PRE-TRAINING DATASETS

Table 9: **Ablation removing one dataset from the ExtendedCHAMMI dataset.** We report the linear evaluation results for DINO BoC.

| Training set | HPA-FOV F1 Protein loc. | HPA-FOV F1 Cell type | Accuracy on JUMP-CP | WTC F1 Cell cycle st. |
|---|---|---|---|---|
| ExtendedCHAMMI | 61.7 | 91.1 | 45.2 | 90.5 |
| minus WTC | 60.4 $_{-1.3}$ | **91.1** $_{-0.0}$ | 43.7 $_{-1.5}$ | 90.9 $_{+0.4}$ |
| minus Cell painting | 60.2 $_{-1.5}$ | **91.1** $_{-0.0}$ | 44.1 $_{-1.1}$ | 89.8 $_{-0.7}$ |
| minus HPA (FOV, single cell) | 41.7 $_{-20.0}$ | 89.9 $_{-1.2}$ | **46.9** $_{+1.7}$ | **92.3** $_{+1.8}$ |
| minus OpenCell | **60.9** $_{-0.2}$ | 91.0 $_{-0.1}$ | 44.2 $_{-1.0}$ | 90.0 $_{-0.5}$ |

To study the influence of specific pre-training datasets on the performance on others, we remove some pre-training datasets in Table 9 and evaluate the performance on HPA-FOV, JUMP-CP and WTC. As expected, when removing both HPA datasets, the protein localization performance is severely altered, but the cell type classification, a much easier task remains accurate. Not much difference is observed on the HPA tasks when removing the other datasets. In general, removing any dataset hurts the overall performance.

## I   RESULTS ON THE EIGHT CHANNELS OF JUMP-CP

As shown in Table 10, both DINO HA and DINO BoC are flexible, resulting in improved performance when the downstream tasks involves a larger number of channels than at pre-training time. Here, the pre-training dataset contains up to 5 channels, while the models are evaluated with up to 8.

Note that the Channel ViT approach cannot be evaluated on images that contain more channels than the maximum number of channels per image of the pre-training data, since there are no trained channel embeddings for the extra channels.

Table 10: **Mean accuracy on the JUMP-CP dataset with models pre-trained on Extended-CHAMMI with a maximum of five channels.**

| | JUMP-CP 5 channels | JUMP-CP 8 channels |
|---|---|---|
| Channel ViT | 39.5 | ✗ |
| DINO HA | **45.2** | 51.4 |
| DINO BoC | **45.2** | **51.6** |

## J   EXPERIMENTS ON AERIAL IMAGERY

To demonstrate the capability of our DINO BoC approach to obtain useful features for imaging domains other than microcopy, we train and benchmark the performance of DINO BoC on the Meter-ML dataset introduced by Zhu et al. (2022) in Table 11.

The Meter-ML dataset contains images acquired by multiple sensors: four channels for NAIP images at resolution $720 \times 720$, four channels from Sentinel-2 at resolution $72 \times 72$, and lower resolution images from Sentinel-2 (S2) and Sentinel-1 (S1). The task consists in classifying sources of methane emissions in six categories (CAFOs, Landfills, Mines, ProcessingPlants, RefineriesAndTerminals, WWTreatment). Zhu et al. (2022) showed that the NAIP images led to better performance, and S2 could improve the result of one class accuracy. We trained one DINO BoC model on NAIP, one on NAIP and S2 images, with only the highest resolution channels, and one model on all S1, S2 and NAIP channels.

Table 11: **DINO BoC outperforms the state-of-the-art models when using all channels on the Meter-ML dataset.** Top: AUROC of models trained on NAIP and Sentinel data. Bottom: AUROC of models using only NAIP channels at inference.

| Approach | Architecture | Test dataset | Pre-training dataset | mAP |
|---|---|---|---|---|
| Meter-ML Zhu et al. (2022) | DenseNet-121 | NAIP, S2, S1 | NAIP, S2, S1 (85K) | 51.7 |
| LHRS-bot Muhtar et al. (2024) | VLM | NAIP, S2, S1 | LHRS-Align-Recap (1.1M images & text) | 71.8 |
| VHM Pang et al. (2024) | VLM | NAIP, S2, S1 | VersaD (14M images & text) | 72.7 |
| DINO BoC(ours) | ViT-L | NAIP, S1, S2 | Meter-ML NAIP, S1, S2 train (85K) | **76.6** |
| Approach | Architecture | Test dataset | Pre-training dataset | mAP |
| Meter-ML Zhu et al. (2022) | DenseNet-121 | NAIP | NAIP (85K) | 54.8 |
| Supervised Cong et al. (2022) | ViT-L | NAIP | fMoW Sentinel (770K) | 69.7 |
| SatMAE Cong et al. (2022) | ViT-L | NAIP | fMoW Sentinel (770K) | 76.9 |
| ScaleMAE Reed et al. (2023) | ViT-L | NAIP | fMoW RGB (363K) | 78.4 |
| USatMAE Irvin et al. (2023) | ViT-L | NAIP | USAtlas NAIP (3.6M) | **83.7** |
| DINO BoC(ours) | ViT-L | NAIP | NAIP, S1, S2 (85K) | 76.5 |
| DINO BoC(ours) | ViT-L | NAIP | NAIP, S2 (85K) | 81.9 |
| DINO BoC(ours) | ViT-L | NAIP | NAIP (85K) | 82.2 |

Using the exact same normalization and evaluation protocol than for the microscopy benchmarks, of our DINO BoC approach outperforms the state-of-the-art by a large margin when using all channels. Using only NAIP imagery, it is close to state-of-the-art approaches, even though it is pre-trained with much smaller datasets (2 orders of magnitude less than the approach of Irvin et al. (2023)), and does not use remote-sensing specific architectures. This demonstrates the generality of our approach to a wider range of imaging domains.

## K    ABLATION ON FEATURE DIMENSION

In Appendix D.2 we describe how the features are obtained for each of the channel-invariant methods. In particular, joint and independent channel encoding strategies naturally lead to different sized embeddings, since the former results in an image level representation, while the later results in channel level representations.

Let $D$ denote the backbone's token dimension, $K$ the number of channels, and $L$ the number of last layers the CLS tokens are taken from. Then, when using both CLS and channel-wise average pooled patch tokens, the feature size for Channel-ViT is $LD + KD$, for DINO BoC it is $LKD + KD$ and for DINO HA it is $L(1 + K)D + KD$. When only CLS tokens are used, the feature size for Channel-ViT is $LD$, for DINO BoC it is $LKD$ and for DINO HA it is $L(1 + K)D$.

In order to demonstrate that DINO BoC outperforms DINO Channel-ViT due to the quality of the features and not due to its dimension, we consider two setups where both models have the same feature size. Those setups are:

1. For Channel-ViT we use only the CLS token from the last layer; while for DINO BoC we average pool the CLS tokens for each channel. Thus for both models the features are $D$-dimensional.

2. For Channel-ViT we concatenate the CLS token from the last layer to the channel-wise average pooled patch tokens, thus the feature is $D + KD$-dimensional. On the other hand, for DINO BoC we concatenate only the CLS tokens for each channel, resulting in a $KD$-dimensional feature.

The results obtained for those setups on the CHAMMI benchmark are listed in Table 12. In both cases DINO BoC outperforms DINO Channel-ViT.

Table 12: **F1 scores for a linear probe on CHAMMI test set: Ablation with similar embedding size.**

| Model | Feature dimension | Average OOD | | | | WTC | | HPA | | | CP | | | |
|---|---|---|---|---|---|---|---|---|---|---|---|---|---|---|
| | | Mean | WTC | HPA | CP | Task1 | Task2 | Task1 | Task2 | Task 3 | Task1 | Task2 | Task3 | Task4 |
| DINO Channel-ViT | $D$ | 59.8 | 66.9 | **76.7** | 35.9 | 83.1 | 66.9 | **88.2** | **84.9** | **68.4** | 80.5 | 54.5 | 23.3 | 30.0 |
| DINO BoC | $D$ | **65.4** | **86.7** | 67.9 | **41.5** | 89.4 | 86.7 | 82.9 | 79.1 | 56.7 | **83.8** | **61.2** | **26.6** | **36.8** |
| DINO Channel-ViT | $KD$ | 63.2 | 74.2 | **77.9** | 37.4 | 86.4 | 74.2 | **90.2** | **86.2** | **69.7** | 83.3 | 56.5 | 24.4 | 31.3 |
| DINO BoC | $KD$ | **67.9** | **89.2** | 74.9 | **39.7** | **90.5** | **89.2** | 88.3 | 84.7 | 65.0 | **90.5** | **60.5** | **25.8** | **32.7** |

