# OpenReview forum: "Scaling Channel-Invariant Self-Supervised Learning"
_ICLR.cc/2025/Conference — Submitted to ICLR 2025_

### Official Review · Reviewer_NZMa · 2024-10-16

**Soundness:** 4
**Presentation:** 4
**Contribution:** 2
**Rating:** 5
**Confidence:** 3

**Summary:**

Some applications of ML (such as biology, medicine, remote sensing) feature different channels across datasets. This channel robustness challenge is under-explored in ML since natural imagery applications do not often face this challenge — they consistently use 3 channels (RGB). This paper focuses on fluorescent microscopy images to investigate channel robustness. (I'm using channel robustness to mean generalizing to missing channels and new channels.) A prior work, Channel-ViT tokenizes channels separately and can be robust to missing channels (similar to remote sensing models like SatMAE and Presto). This paper introduces a ViT architecture called DINO-HA. This paper thoroughly benchmarks Channel-ViT, DINO-HA, and DINO-BoC (Bag of channels) using large backbones (ViT-L) and large pretraining datasets. This paper finds that DINO-BoC often outperforms Channel-ViT, evaluated as a frozen feature extractor.

**Strengths:**

- This paper is well written
- Extensive experiments are performed. Specifically, using ViT-Large models, two different SSL algorithms, different pretraining datasets, and several in-distribution and out-of-distribution test sets.
- From my reading, the core finding is that a "bag of channels" approach can often outperform models that allow inter-channel reasoning. This is interesting and surprising.

**Weaknesses:**

- Channel-ViT (the baseline) beats DINO-HA on 17 of 26 test sets on page 8. And DINO-BoC beats Channel-ViT on 16 of 26 test sets on page 8. I'm not convinced that DINO-HA is valuable to the community beyond these fairly negative results.
- I do not consider DINO-BoC to be a novel method. It process channels separately using a ViT, then concatenates embeddings.

I recognize this is an applied paper and the bar for ML novelty is lower, however I question the value to the broader ICLR community. That being said, the experiments are extensive and the findings are interesting.

**Questions:**

As far as I can tell, DINO-BoC concatenates channel embeddings to form the full-sample embedding using the CLS tokens. This embedding is a D * N-dimensional vector (where D is the ViT width and N is the number of channels). And the full-sample embedding of Channel-ViT is a D-dimensional vector, taken from the CLS token. Although in the appendix it says the CLS token is concatenated across the last 4 layers for linear probing.

Can you please clarify how you build full-sample embeddings for each of the three methods? If Channel-ViT's embeddings are indeed D-dimensional and DINO-BoC are D * N-dimensional, I'd like to see results when they have the same sized embeddings. This can be done by taking the mean between channels (for DINO-BoC) to achieve D-dimensional embeddings. This can be done by concatenating the mean-pooled patch representations for Channel-ViT to achieve N * D-dimensional embeddings (specifically, you'd pool the patches for each channel, then concatenate these pooled channel-wise embeddings).

**Details Of Ethics Concerns:**

I have no ethics concerns.

---

> ### Author Response · Authors · 2024-11-24
>
> Thank you for taking the time to review our paper! Your question regarding the novelty of our method was raised by other reviewers, in addition to our comments below, please also see the general response section.
>
> > I do not consider DINO-BoC to be a novel method. It process channels separately using a ViT, then concatenates embeddings.
>
> Addressing your concerns specifically, our main contribution is that, contrary to recent works that argued in favor join encoding strategies (Bourriez et al 2024, Bao et al 2024, Kraus et al 2024, Pham & Plummer 2024), we find that (surprisingly) at scale the winning channel-invariant strategy is to combine independent channel encoding (BoC approach) with ViTs! We think this is rather striking result given that (a) Xun et al 2024, in their Fig.S1, found that ViTs underperformed CNNs when training on single channels (BoC approach) and (b) Bouriez et al 2024, in their Fig.4, reported that the BoC approach in turn underperforms channel-adaptive ViTs.
>
> > Channel-ViT (the baseline) beats DINO-HA on 17 of 26 test sets on page 8. And DINO-BoC beats Channel-ViT on 16 of 26 test sets on page 8. I'm not convinced that DINO-HA is valuable to the community beyond these fairly negative results.
>
> To your question about the value of the Hierarchical Attention model, it achieves a trade-off between jointly encoding the channels (Channel-ViT) and independently encoding them (BoC strategy), as such, it offers an intermediate point of analysis between joint and independent channel encoding. In our work, we find that inter-channel reasoning is detrimental to the performance and generalization capability of channel-invariant models. Therefore, the HA model results are pertinent because it strongly outperforms Channel-ViT, while not reaching the performance of BoC, supporting the conclusion that moving from joint to independent channel encoding is indeed the most promising direction for channel-invariant models. Please also see our response to reviewer TY5V regarding our HA model.

---

> > ### Comment · Reviewer_NZMa · 2024-11-24
> >
> > Thanks for the rebuttal and the additional experiments presented in your general comments.
> >
> > I appreciate, but am not convinced by the remote sensing experiments. DINO BoC differs from SatMAE and ScaleMAE w.r.t. pretraining data and pretraining method — both of which greatly impact the learned representations, and especially when evaluating them as frozen feature extractors. Shouldn't you swap out SatMAE's architecture for your BoC ViT and hold everything else constant?

---

> > > ### Author Response · Authors · 2024-11-26
> > >
> > > Thank you for spending time to review our answers and for this additional question! We agree that the pre-training data and SSL method greatly influence the learned representations, however, the pre-training method, like the architecture, is an integral component of the feature encoder, which is why we do not keep it constant. Regarding the pre-training data, yes, we will also compare DINO BoC to SatMAE and Scale-MAE using the same pre-training data (fMoW Sentinel and fMoW RGB respectively), while evaluating on NAIP. We are still training the DINO BoC model on fMoW and will present results along with SatMAE and Scale-MAE in a separate table as transfer-learning experiments. These results may not be ready in time for the rebuttal, but will be included in the future version of the work.
> > >
> > > Nevertheless, we think the comparison to SatMAE and Scale-MAE (both ViT-L) is valuable even now because both models achieved SOTA results at the time on NAIP, even compared to supervised ViT-L models trained on NAIP from scratch. We update our table to include this baseline, as well as indications as to the architecture. In addition, the USatMAE (also ViT-L) model was not only trained on the 3.6M images of USAtlas -- which enriches NAIP by matching Sentinel-2 images to it and applying various up- and down-sampling strategies -- but also fine-tuned on the Meter-ML NAIP dataset we use.  As such, our comparisons clearly demonstrate that DINO BoC generalizes to other relevant non-microscopy domains. Beyond this, they show that DINO BoC achieves close to SOTA results on concrete problems using vastly less pre-training data, out of the box.
> > >
> > > | **Approach**       | **Architecture** | **Test dataset** | **Pre-training dataset**                       | **mAP**  |
> > > |---------------------|------------------|------------------|-----------------------------------------------|----------|
> > > | Meter-ML [Zhu et al., 2022] | DenseNet-121      | NAIP, S2, S1      | NAIP, S2, S1 (85K)                           | 51.7     |
> > > | LHRS-bot [Muhtar et al., 2024] |    Custom VLM              | NAIP, S2, S1      | LHRS-Align-Recap (1.1M images & text)       | 71.8     |
> > > | VHM [Pang et al., 2024]       |       Custom VLM           | NAIP, S2, S1      | VersaD (14M images & text)                  | 72.7     |
> > > | **DINO BoC** (ours)     | ViT-L            | NAIP, S1, S2     | Meter-ML NAIP, S1, S2 (85K)           | **74.4** |
> > >
> > > | **Approach**       | **Architecture** | **Test dataset** | **Pre-training dataset**                     | **mAP**  |
> > > |---------------------|------------------|------------------|-----------------------------------------------|----------|
> > > | Meter-ML [Zhu et al., 2022] | DenseNet-121      | NAIP             | NAIP (85K)                                  | 54.8     |
> > > | Supervised [Cong et al., 2022]  | ViT-L             | NAIP             | fMoW Sentinel (770K)                        | 69.65     |
> > > | SatMAE [Cong et al., 2022]  | ViT-L             | NAIP             | fMoW Sentinel (770K)                        | 76.9     |
> > > | ScaleMAE [Reed et al., 2023]| ViT-L             | NAIP             | fMoW RGB (363K)                             | 78.4     |
> > > | USatMAE [Irvin et al., 2023]| ViT-L             | NAIP             | USAtlas NAIP (3.6M)                         | 83.7     |
> > > | **DINO BoC** (ours)       | ViT-L            | NAIP             | NAIP (85K)                                  | 82.2     |
> > > | **DINO BoC** (ours)      | ViT-L            | NAIP             | NAIP, S2 (85K)                              | 81.9     |

---

> ### Author Response · Authors · 2024-11-26
>
> Regarding your questions about the features' dimension:
>
> > As far as I can tell, DINO-BoC concatenates channel embeddings to form the full-sample embedding using the CLS tokens. This embedding is a D * N-dimensional vector (where D is the ViT width and N is the number of channels). And the full-sample embedding of Channel-ViT is a D-dimensional vector, taken from the CLS token.
> >
> > Although in the appendix it says the CLS token is concatenated across the last 4 layers for linear probing.
> >
> >Can you please clarify how you build full-sample embeddings for each of the three methods?
>
> We have updated Appendix G.2 (now Appendix D.2) to more clearly describe how the features are obtained for each of the methods. Let:
> - $D$: embedding dimension
> - $K$: number of channels
> - $L$: number of “last” layers we take the CLS tokens from
>
> For all evaluations -- with the exception of the evaluations on the CHAMMI benchmark -- we take $L=4$ and the features are obtained by concatenating the CLS tokens across the last $L$ layers, as well as the channel-wise average pooled patch tokens. In more details:
> - For Channel-ViT we concatenate the CLS tokens from the last L layers and the channel-wise average pooled patch tokens, so $LD + KD$ dimensional.
> - For BoC we concatenate the CLS tokens, for each channel, from the last L layers and the channel-wise average pooled patch tokens, so $LKD + KD$ dimensional.
> - For HA we concatenate the global and channel CLS tokens from the last L layers and the channel-wise average pooled patch tokens, so $L(1+K)D + KD$ dimensional.
>
> For the evaluations on the CHAMMI benchmark, only the CLS tokens are used and $L=1$. Therefore the feature size for Channel-ViT is $D$; for BoC it is $KD$ and for HA it is $(1+K)D$.
>
> > If Channel-ViT's embeddings are indeed D-dimensional and DINO-BoC are D * N-dimensional, I'd like to see results when they have the same sized embeddings.
> >
> > This can be done by taking the mean between channels (for DINO-BoC) to achieve D-dimensional embeddings.
> >
> > This can be done by concatenating the mean-pooled patch representations for Channel-ViT to achieve N * D-dimensional embeddings (specifically, you'd pool the patches for each channel, then concatenate these pooled channel-wise embeddings).
>
> Thanks for this suggestion! We are adding those two comparisons to Appendix K. We present the results for the following two scenarios:
> 1. For Channel-ViT we use only the CLS token from the last layer, so $D$ dimensional; while for BoC we average pool the CLS tokens for each channel, so $D$ dimensional as well.
> 2. For Channel-ViT we concatenate the CLS token from the last layer to the channel-wise average pooled patch tokens, so $D + KD$ dimensional; while for BoC we concatenate only the CLS tokens for each channel, so $KD$ dimensional.
>
> The results we obtained on the CHAMMI benchmark for those configurations are:
> | **Model**         | **Feature dimension** | **Mean** | **WTC** | **HPA** | **CP** | **WTC Task1** | **WTC Task2** | **HPA Task1** | **HPA Task2** | **HPA Task3** | **CP Task1** | **CP Task2** | **CP Task3** | **CP Task4** |
> |--------------------|-----------------------|----------|---------|---------|---------|---------------|---------------|---------------|---------------|---------------|---------------|---------------|---------------|---------------|
> | **DINO Channel-ViT** | $D$              | 59.8     | 66.9    | **76.7**| 35.9    | 83.1          | 66.9          | **88.2**      | **84.9**      | **68.4**      | 80.5          | 54.5          | 23.3          | 30.0          |
> | **Ours**           | $D$              | **65.4** | **86.7**| 67.9    | **41.5**| **89.4**          | **86.7**      | 82.9          | 79.1          | 56.7          | **83.8**      | **61.2**      | **26.6**      | **36.8**      |
> | **DINO Channel-ViT** | $KD$            |    63.2      | 74.2    |    **77.9**     |   37.4      | 86.4          | 74.2          | **90.2**      | **86.2**      |     **69.7**          | 83.3          | 56.5          | 24.4          |        31.3       |
> | **Ours**           | $KD$             | **67.9** | **89.2**| 74.9    | **39.7** | **90.5**      | **89.2**      | 88.3          | 84.7          | 65.0          | **90.5**          | **60.5**          | **25.8**          | **32.7**      |
>
> From those partial results, we observe that the superior performance of BoC is indeed due to the better quality of the features that are learned, rather than a consequence of a larger embedding size.
>
> Still, as the ChannelViT $KD$ results are a bit better than the ChannelViT $D$ results, we will update the other CHAMMI benchmark tables of the paper with this baseline which is more fair.

---

> > ### Comment · Reviewer_NZMa · 2024-11-26
> >
> > Thanks for these experiments and for clarifying how the full-sample embeddings are built.
> >
> > >We agree that the pre-training data and SSL method greatly influence the learned representations, however, the pre-training method, like the architecture, is an integral component of the feature encoder, which is why we do not keep it constant. Regarding the pre-training data, yes, we will also compare DINO BoC to SatMAE and Scale-MAE using the same pre-training data (fMoW Sentinel and fMoW RGB respectively), while evaluating on NAIP. We are still training the DINO BoC model on fMoW and will present results along with SatMAE and Scale-MAE in a separate table as transfer-learning experiments. These results may not be ready in time for the rebuttal, but will be included in the future version of the work.
> >
> > I'm still not convinced by this answer. As far as I can tell, the main contribution of this work is the empirical finding that BoC works surprisingly well for microscopy images. This is primarily demonstrated by comparing DINO-Channel-ViT with DINO-BoC — fair enough. But when trying to demonstrate the effectiveness on remote sensing images, there should be a comparison between DINO-Channel-ViT or DINO with the SatMAE model architecture and the proposed DINO-BoC.
> >
> > Today is the last day of the discussion period, so I will make my decision to keep the score a 5. I do not think the paper is ready to be published at ICLR 2025 — primarily because of its limited methodological novelty.

---

> > > ### Author Response · Authors · 2024-12-04
> > >
> > > Thank you again for reviewing our answers! Following your suggestions, we have additionally trained and evaluated DINO Channel-ViT on aerial imagery. Note that we use the same architecture (ViT-L) as SatMAE, ScaleMAE and USatMAE.
> > >
> > > | Approach                  | Architecture | Test Dataset   | Pre-training Dataset                               | mAP  |
> > > |---------------------------|--------------|----------------|---------------------------------------------------|-------|
> > > | Meter-ML [Zhu et al. 2022]| DenseNet-121 | NAIP, S2, S1   | NAIP, S2, S1 (85K)                                | 51.7  |
> > > | LHRS-bot [Muhtar et al. 2024] | VLM          | NAIP, S2, S1   | LHRS-Align-Recap (1.1M images & text)            | 71.8  |
> > > | VHM [Pang et al. 2024]    | VLM          | NAIP, S2, S1   | VersaD (14M images & text)                       | 72.7  |
> > > | DINO Channel ViT          | ViT-L        | NAIP, S1, S2   | Meter-ML NAIP, S1, S2 train (85K)                | 67.5  |
> > > | DINO BoC                 | ViT-L        | NAIP, S1, S2   | Meter-ML NAIP, S1, S2 train (85K)                | **76.6** |
> > >
> > >
> > >
> > > | Approach                  | Architecture | Test Dataset   | Pre-training Dataset                               | mAP  |
> > > |---------------------------|--------------|----------------|---------------------------------------------------|-------|
> > > | Meter-ML [Zhu et al. 2022]| DenseNet-121 | NAIP           | NAIP (85K)                                       | 54.8  |
> > > | Supervised [Cong et al. 2022] | ViT-L        | NAIP           | fMoW Sentinel (770K)                             | 69.7  |
> > > | SatMAE [Cong et al. 2022] | ViT-L        | NAIP           | fMoW Sentinel (770K)                             | 76.9  |
> > > | ScaleMAE [Reed et al. 2023] | ViT-L        | NAIP           | fMoW RGB (363K)                                  | 78.4  |
> > > | USatMAE [Irvin et al. 2023]| ViT-L        | NAIP           | USAtlas NAIP (3.6M)                              | **83.7** |
> > > | DINO Channel ViT          | ViT-L        | NAIP           | NAIP, S1, S2 (85K)                               | 60.7  |
> > > | DINO Channel ViT          | ViT-L        | NAIP           | NAIP (85K)                                       | 68.6  |
> > > | DINO BOC                  | ViT-L        | NAIP           | NAIP, S1, S2 (85K)                               | 76.5  |
> > > | DINO BoC                 | ViT-L        | NAIP           | NAIP, S2 (85K)                                   | 81.9  |
> > > | DINO BoC               | ViT-L        | NAIP           | NAIP (85K)                                       | 82.2  |
> > >
> > >
> > > Following the same trend observed for microscopy images, DINO BoC strongly outperforms DINO Channel-ViT, demonstrating that DINO BoC holds promise for imaging domains other than microscopy.
> > >
> > > It is notable that we used the exact same normalization and evaluation protocol as for the microscopy benchmarks, no fine-tuning was performed. Furthermore, compared to SatMAE, ScaleMAE, USatMAE, LHRS-bot and VHM, our models are trained on a significantly smaller dataset, which puts them at a disadvantage. Even so, DINO BoC outperforms the sota on the NAIP+S1+S2 evaluation, and is close to it on the NAIP evaluation.
> > >
> > >  This highlights the out-of-the-box generality of our approach across a wider range of imaging domains.

---

### Official Review · Reviewer_TY5V · 2024-10-25

**Soundness:** 2
**Presentation:** 2
**Contribution:** 1
**Rating:** 3
**Confidence:** 3

**Summary:**

The paper presents a large set of SSL experiments in the microscopy imaging domains and scales existing methods with the existing DINOv2 approach for channel-variant images.

**Strengths:**

The paper addresses an important problem related to channel-varying images in microscopy imaging and conducts an extensive set of experiments, including OOD generalization, within this domain. The trained models generally outperform the existing literature.

**Weaknesses:**

1) The paper's title, "Scaling Channel-Invariant Self-Supervised Learning," suggests broad applicability across various imaging domains. However, the experiments are limited to fluorescent microscopy images despite the relevance of channel-invariant features in other fields, such as medical and satellite imagery, as also acknowledged by the authors in the abstract. Prior work on channel-invariant learning has demonstrated applications across diverse image domains [1], including satellite imaging. A broader experimental scope of imaging domains (e.g., see [1]), rather than larger datasets on a single one, would align the paper’s claims with the actual generality and applicability of channel-invariant self-supervised learning across relevant fields.

2) Contribution (i), which involves selecting DINOv2 and integrating it with the Bag of Channels approach (an existing SSL algorithm combined with a known channel-invariant module), is challenging to consider a meaningful contribution on its own since the combination of these methods lacks substantial technical novelty, as it doesn’t introduce new mechanisms or address specific challenges unique to this approach.

3) The technical novelty (contribution (ii)) is limited, as the proposed Hierarchical Attention module appears to be a combination of existing methods (Channel-ViT and Bag of Channels) with the addition of extra "channel" tokens and corresponding attention masking. Furthermore, the clarity of presentation of the technical contribution needs improvement, as I explained in the next point. It is also unclear to me the motivation behind introducing DINO HA since, in Tables 1, 2, and 3, the best method is usually DINO Bag of Channels and not DINO Hierarchical Attention. What is the purpose of proposing two methods, with one clearly better than the other?

4) The technical presentation of contributions (i) and (ii) is somewhat unclear and misleading. More precisely, the main technical section 3, which is named "Problem Formulation" (renaming needed), should instead focus on the description of the introduced modules and clearly discern the contributions of the paper (sections 3.3 (?) and 3.4) against the already existing strategies (i.e., sections 3.1, 3.2, 3.5). To clarify the contributions, it would help to split Section 3 into two sub-sections (Preliminaries vs Methodology) to help the reader better grasp the actual contributions. I would also suggest a better design of Figure 2, e.g., in a single panel, highlighting the differences between previous and proposed work.

5) The related work section concerning SSL with microscopy images and section 4.1 are comprehensive but verbose and would need some summarization and re-writing.

[1] Bao et al., Channel Vision Transformers: An Image Is Worth 1 x 16 x 16 Words, ICLR 2024

**Questions:**

Unfortunately, while the paper tackles a relevant problem and demonstrates better performance through a large set of experiments, the contribution is limited by a lack of generalizability across (untested) imaging domains and, most importantly, by insufficient technical novelty.

See the weaknesses section to improve the technical presentation of the contributions. I am open to hearing the feedback from the authors on the rebuttal period.

---

> ### Author Response · Authors · 2024-11-24
>
> Thank you for your detailed comments and suggestions! Your concerns regarding the novelty of our work and applicability across various imaging domains (your cited weaknesses 1 & 2) were shared by other reviewers, we hence address them in the general response section.
>
> > The technical novelty (contribution (ii)) is limited, as the proposed Hierarchical Attention module appears to be a combination of existing methods (Channel-ViT and Bag of Channels) with the addition of extra "channel" tokens and corresponding attention masking. Furthermore, the clarity of presentation of the technical contribution needs improvement, as I explained in the next point. It is also unclear to me the motivation behind introducing DINO HA since, in Tables 1, 2, and 3, the best method is usually DINO Bag of Channels and not DINO Hierarchical Attention. What is the purpose of proposing two methods, with one clearly better than the other?
>
> Regarding the Hierarchical Attention (HA) model, let us separately address the two concerns that were raised: its technical novelty, and its purpose.  With respect to its novelty, it achieves a trade-off between jointly encoding the channels (Channel-ViT) and independently encoding them (Microsnoop and BoC strategy). Yet, it is not a simple combination. HA is technically distinct from both methods.
>
> The only common aspect between Channel-ViT and HA is that they employ single-channel patches. However, Channel-ViT differentiates between tokens coming from different patches through channel embeddings, and there is no restriction on cross-channel attention. On the other hand, HA employs a novel attention scheme that only allows cross-attention between patch tokens belonging to a same channel, then cross-channel interactions are restricted to the global CLS token that attends to the channel CLS tokens. This way, the HA model offers both independent channel encodings through the channel CLS tokens, and an image-level encoding, through the global CLS token.
>
> In regards to the purpose of the HA model, it offers an intermediate point of analysis between joint and independent channel encoding. In our work, we find, surprisingly, that at scale inter-channel reasoning is detrimental to the performance and generalization capability of channel-invariant models. Therefore, the HA model results are relevant because it strongly outperforms Channel-ViT, although not reaching the performance of BoC. This counters the consensus of a series of recent papers and suggests instead that moving from joint to independent channel encoding is the most promising direction to train foundation models in domains with variable input channels.
>
> > The technical presentation of contributions (i) and (ii) is somewhat unclear and misleading. More precisely, the main technical section 3, which is named "Problem Formulation" (renaming needed), should instead focus on the description of the introduced modules and clearly discern the contributions of the paper (sections 3.3 (?) and 3.4) against the already existing strategies (i.e., sections 3.1, 3.2, 3.5). To clarify the contributions, it would help to split Section 3 into two sub-sections (Preliminaries vs Methodology) to help the reader better grasp the actual contributions. I would also suggest a better design of Figure 2, e.g., in a single panel, highlighting the differences between previous and proposed work.
> >
> > The related work section concerning SSL with microscopy images and section 4.1 are comprehensive but verbose and would need some summarization and re-writing.
>
> Thank you for the suggestions regarding the structure of the paper, we have made broad revisions to the text that we believe significantly clarify the contributions and results of our work.
>
> Please let us know if we have addressed your questions, and if you have any additional suggestions.

---

> > ### Comment · Reviewer_TY5V · 2024-11-25
> >
> > I would like to thank the authors for their rebuttal and the corresponding explanations regarding the points I raised.
> >
> > I appreciate the re-framing of the title, given that all the experiments come from microscopy images, but I don't like the addition of experiments from a completely different domain after the change of scope (Appendix J).
> >
> > I think that the clarity of the paper is improved after re-factoring Sections 2 and 3. I also understood, after the detailed rebuttal, that the HA approach conceptually lies between ChannelViT and BoC. Furthermore, the authors also clarified that their contribution lies in a "different way" of doing independent-channel encoding via either BoC (at the extreme) or in the middle via HA. I still do not like the idea of proposing both BoC and HA since BoC is practically more effective. HA only complicates the modeling for doing independent-channel encoding "inside the architecture", while BoC reaches the same goal with a simpler design. The authors would achieve the same goal of showing the superiority of independent-channel architectures without adding redundant concepts. Also, the current title does not really point the reader to the claimed novelty derived from the superiority of independent-channel modeling. Therefore, I strongly believe that the authors should re-frame the presentation of their technical contribution, from the title to the main proposed architecture and corresponding ablation studies.
> >
> > Apart from the above concerns, my more significant concern remains the novelty of the proposed two contributions, BoC in particular. The same conceptual idea of encoding independent channels is indeed proposed by Microsnoop. The only difference between the proposed BoC and Microsnoop is that they use different SSL objectives (DINO vs. MAE) and architecture (ViT vs. CNN). Therefore, the paper does not introduce any novel conceptual idea but instead adapts an existing approach to a different architecture. In practice, the SSL objective does not even need any particular adaptation. I am afraid that this technical contribution does not reach the bar for ICLR papers.

---

> > > ### Author Response · Authors · 2024-11-26
> > >
> > > Hi TY5V - thank you for getting back to us and for your time. We really appreciate it!
> > >
> > > As you point out, while DINO BoC and HA are different to Xun et al 2024 in both architecture and SSL objective, the value of our contribution to the ICLR and broader research community lies not in its technical novelty. Instead, it is a scientific challenge to the conclusions of previous published works: against all previous evidence, we present for the first time **empirical results** showing that - at scale - a conceptually simple method, when combined with ViT and DINO, outperforms all other methods that implement more sophisticated priors and bespoke (novel) techniques tailored to the problem setting. Our results also pose new **theoretical questions**, as the results demonstrate for the first time that the assumption that joint channel-encoding is beneficial in non-RGB, variable-channel domains, does not hold at scale, and is even detrimental. Both aspects are further substantiated by our (control) experiments on DINO HA, incidentally a technically novel and innovative method that, while outperforming all previous methods, surprisingly still falls behind fully independent encoding of channels.
> > >
> > > As the field is poised to train very large cross-dataset models for channel-heterogeneous scientific imaging domains in the near future, our results also provide critical **guidance for practitioners**: after reading our paper, which approach would you choose ? What about before?
> > >
> > > Clearly, technical and conceptual novelty of methods can be important, but it is not the only criterion for the value of a contribution to the ICLR and broader scientific community. Our paper is an applied work. It presents surprising (novel) empirical results, challenges theoretical assumptions, provides novel practitioner guidance, and establishes a **new state-of-the-art** on this - as you acknowledge - important problem. We think these factors too should be accounted for in the contribution.
> > >
> > > On top, we specifically address your cited weakness regarding lack of data that clearly demonstrates that our findings generalize beyond microscopy images, which should only strengthen your evaluation. Still, we appreciate your remaining concerns about the presentation of our findings. Thank you again your continued feedback and for your patience while we work on further revisions for the final version, along with even more imminent experimental results.

---

### Official Review · Reviewer_6FWq · 2024-11-03

**Soundness:** 2
**Presentation:** 1
**Contribution:** 2
**Rating:** 5
**Confidence:** 5

**Summary:**

The paper proposes a self-supervised learning method for heterogeneous multi-channel images. The proposed method uses DINO v2 as a backbone and integrates bag of channels and hierarchical attention into Channel ViT. The work is evaluated on microscopy images extensively.

**Strengths:**

1. The experiments on microscopy datasets are extensive and the reported numbers provide empirical understandings.

**Weaknesses:**

Major Concerns:
1. The title describes a generic method for multi-channel images, but the paper focuses only on microscopy images. Considering that different multi-channel images may exhibit different behaviors, the title seems to be a bit big. The authors shall either broaden their experiments to include other types of multi-channel images, or narrow the title and framing to focus specifically on microscopy images.
2. The title states this work focuses on self-supervised learning. However, after reading the whole paper, the method is not generic. On the contrary, it seems to be designed based on DINO v2. No discussion on the choice of DINO v2 at all.  The authors shall provide a clear justification for choosing DINO v2 as their base method. Additionally, please discuss how your approach might generalize to other self-supervised learning frameworks, or explain why your method is specific to DINO v2.
3. The contributions on self-supervised learning is not clear to me. In my opinion, the paper seems to be a multi-channel variant of DINO v2, which is an incremental work. But, the paper tries to tell a different story. If this work is about self-supervised learning while leveraging multi-channel images, please specify why this work differs from studies of self-supervised learning on RGB images. If this work focuses on adapting existing self-supervised learning works on multi-channel images, please clarify it from the title to introduction.
4. The technical writing is a bit hard to follow. The presentation of the method proposed is unclear. The paper spends quite some space the background while the methodological details are provided in the appendix. It is recommended to reorganize the related works and method sections to show key methodological results.

**Questions:**

1. How does this method work on multi-channel images such as remote sensing or multi-spectral images?
2. Does this method work on other self-supervised learning framework?
3. Follow the same logic, will better self-supervised learning backbone show better results?

---

> ### Author Response · Authors · 2024-11-24
>
> Thank you so much for taking the time to review our paper! Many of your concerns were shared by other reviewers. We specifically comment on your major concerns below, but please also see our general response section.
>
> > The title describes a generic method for multi-channel images, but the paper focuses only on microscopy images. Considering that different multi-channel images may exhibit different behaviors, the title seems to be a bit big. The authors shall either broaden their experiments to include other types of multi-channel images, or narrow the title and framing to focus specifically on microscopy images.
>
> We agree that our focus on the domain of microscopy images should be reflected in the title and we have narrowed it. Nevertheless, our method is indeed generic, and we now show with new experiments that it works very well out of the box for aerial and remote sensing imagery (see general responses and Appendix J).
>
> > The title states this work focuses on self-supervised learning. However, after reading the whole paper, the method is not generic. On the contrary, it seems to be designed based on DINO v2. No discussion on the choice of DINO v2 at all. The authors shall provide a clear justification for choosing DINO v2 as their base method. Additionally, please discuss how your approach might generalize to other self-supervised learning frameworks, or explain why your method is specific to DINO v2.
> >
> > The contributions on self-supervised learning is not clear to me. In my opinion, the paper seems to be a multi-channel variant of DINO v2, which is an incremental work. But, the paper tries to tell a different story. If this work is about self-supervised learning while leveraging multi-channel images, please specify why this work differs from studies of self-supervised learning on RGB images. If this work focuses on adapting existing self-supervised learning works on multi-channel images, please clarify it from the title to introduction.
>
> To your questions on self-supervised learning (SSL): as the title suggests, our paper is not concerned with SSL per se, but rather with the scaling properties of “channel-invariant” SSL. In particular, contrary to RGB images, many scientific imaging modalities, such as microscopy images, vary in number and content of channels (see Fig. 1); e.g. WTC-11 has 3 channels, HPA-FOV has 5. Such datasets are also highly heterogeneous with respect to available labels. To explore scaling of channel-invariant methods, SSL is hence a natural choice.
>
> Indeed, we show that our method is compatible with any SSL framework, not just DINOv2. For instance in Table 5 of the original manuscript we compare the performance of BoC pre-trained with DINOv2 and MAE, the dominant paradigm for microscopy foundational models. Noticeably, BoC pre-trained with MAE still performs on par or better than Channel-ViT. Still, we find that DINOv2 substantially outperforms MAE, even when handicapped by a smaller model size. We present these results more prominently now, in a revised Table 1 (see also general responses).
>
> Yet, our main objective is to study the behavior of different channel-invariant strategies at scale. Contrary to findings by several high-profile papers that have explored channel-invariant learning strategies at smaller scale (Bourriez et al 2024, Bao et al 2024, Kraus et al 2024, Pham & Plummer 2024), and noticeably, in further stark contrast to established best-practices for RGB images, encoding channels independently is the winning strategy for large-scale channel-invariant learning with SSL. We thus empirically refute previous claims about the importance of joint channel encoding and, with DINO BoC, present the best performing channel-invariant model for microscopy to date.
>
> > The technical writing is a bit hard to follow. The presentation of the method proposed is unclear. The paper spends quite some space the background while the methodological details are provided in the appendix. It is recommended to reorganize the related works and method sections to show key methodological results.
>
> We have revised structure, writing, tables (Tables 1-2), and figures (Fig. 2), as well as added additional experiments that together, we think, substantially strengthen the presentation.
>
> > Does this method work on other self-supervised learning framework? Follow the same logic, will better self-supervised learning backbone show better results?
>
> As described above, BoC is compatible with any SSL method. We compare DINOv2 to MAE.
>
> > How does this method work on multi-channel images such as remote sensing or multi-spectral images?
>
> As mentioned above and in the general response section, we demonstrate Appendix J that our method generalizes very well to aerial and remote sensing imagery, out of the box.

---

> > ### Comment · Reviewer_6FWq · 2024-11-26
> > **Efforts acknowledged.**
> >
> > I can see the efforts shown in the rebuttal where the authors have utilized the limited rebuttal period. Thus, I decide to slightly upgrade my score.
> >
> > Yet, the paper still has enough space to be improved. I would suggest narrowing down the scope and making this work a concrete one. The current version is still incomplete.
> >
> > 1. Additional experiments are still encouraged. I appreciate that the authors have tried to address my questions with a few experiments.
> > 2. I would suggest reorganize the paper for better presentation. Changing the title does not solve everything.

---

> > > ### Author Response · Authors · 2024-12-04
> > >
> > > Thank you again for your time and for your feedback! It’s been very helpful and we’re glad that our improvements were meaningful in your eyes. We have now even further polished the paper. We wonder if you could give it one more glance with the following points in mind:
> > >
> > > > I would suggest reorganize the paper for better presentation. Changing the title does not solve everything.
> > >
> > > Regarding the paper organization, we made multiple adjustments beyond narrowing the scope of the title, addressing the points you raised in your original review. We streamlined the “Introduction” and “Related Work” section. The “Method” section was reformulated to convey more clearly the contrast between joint and independent encoding strategies, and how our models differ from previous ones. Also the technical novelty and value of the Hierarchical Attention model is emphasized.
> > > We have also significantly changed the flow of the “ Experiments” section, in order to highlight our main contribution, i.e.,  against all previous evidence, we present for the first time empirical results showing that - at scale - the strategy of independently encoding the channels, when combined with ViT and DINO, outperforms all other methods that implement more sophisticated joint encoding strategies. The comparison between different SSL methods features more proeminently than before. Furthermore, we have added experimental results directly comparing DINO BoC to Microsnoop on the Cyclop dataset -- on which Microsnoop reported the largest gains -- demonstrating the DINO BoC outperforms Microsnoop.
> > >
> > > > Additional experiments are still encouraged. I appreciate that the authors have tried to address my questions with a few experiments.
> > >
> > > We have either added new experimental result or featured more clearly existing results addressing your two previous questions:
> > > - on whether other self-supervised learning frameworks show better results (Table 1);
> > > - and on how this method works on multi-channel images such as remote sensing or multi-spectral images (Table 11).

---

### Official Review · Reviewer_PzT7 · 2024-11-04

**Soundness:** 3
**Presentation:** 4
**Contribution:** 3
**Rating:** 6
**Confidence:** 3

**Summary:**

This work study different channel invariant strategies to train vision transformers for self supervised learning. These approaches are study in the context of fluorescent microscopy images where different datasets carry different combination of channels. Channel invariance allows for self-supervised pretraining using a combination of all existing datasets which translates to better pretrained models. Authors also proposed hierarchical attention as a novel way to train channel invariant models.

**Strengths:**

* Paper is well written and technically sound
* The work address a significant problem for applying machine learning to microscopy images at scale
* Proper evaluation of the different approaches is coducted

**Weaknesses:**

* It is not clear how the bag of channels approach claimed as yours is different to Microsnoop form Xun et al.
* The proposed hierarchical attention approach rarely beat the bag of channel approach
* Work only tested on a single application domain.

**Questions:**

* Do you expect this approach to work for other application domains with image modalities of different number of channels with limited channels matching across datasets like it is often the case in remote sensing?
* Figure 2 legend can be improved by adding more details to it.

---

> ### Author Response · Authors · 2024-11-24
>
> Thank you very much for your review! Your questions regarding how our method is different from Microsnoop and whether it works on other domains was shared by other reviewers. In addition to our comments below, please also see the general response section.
>
> > It is not clear how the bag of channels approach claimed as yours is different to Microsnoop form Xun et al.
>
> Regarding the difference between DINO Bag of Channels (BoC) and Microsnoop, there are three main components to be considered in a channel-invariant encoder:
> 1. the strategy that technically allows the model to handle channel heterogeneous data;
> 2. the network architecture itself;
> 3. the training strategy.
>
> DINO BoC and Microsnoop share the channel-invariant strategy, i.e., independently encoding each channel, however they differ significantly on the other aspects: DINO BoC leverages a ViT and the DINOv2 SSL method, while Microsnoop consists of a convolutional U-Net pre-trained with a Masked Auto-Encoder (MAE) SSL objective.
>
> We note here that, Xun et al 2024 (see Fig.S1), found that ViTs underperformed convolutional U-Nets when training on BoC and Bouriez et al 2024 (see Fig.4), reported that the BoC approach in turn underperforms channel-adaptive strategies with ViTs. Rather surprisingly then, our results show that combining ViTs with BoC outperforms all other methods at scale!
>
> In Table 5 of the original manuscript (now featured more prominently in Table 1), we show that, for BoC, the choice of DINOv2 further yields significant improvements over MAE, that is, the SSL learning objective used in Microsnoop.
>
> Finally, we added a new comparison to Microsnoop, evaluating our approach on the Cyclops dataset, which is highly out of domain in our case, in which DINO BoC outperforms Microsnoop by a large margin (8 points on average; up to 30 points on rare classes).
>
> > The proposed hierarchical attention approach rarely beat the bag of channel approach
>
> With respect to the Hierarchical Attention (HA) approach, its main value is providing an intermediate point of comparison between the joint and independent encoding of channels. By using a hierarchical attention scheme that limits the inter-channel interactions, it outperforms Channel-ViT, while complete independent encoding (BoC) is the best approach at scale, contrary to previous findings at small scale (Bourriez et al 2024). Therefore, the performance of the HA model further corroborates our finding that moving from joint to independent channel encoding is the most promising direction to train foundation models in domains with variable input channels.
>
> > Do you expect this approach to work for other application domains with image modalities of different numbers of channels with limited channels matching across datasets like it is often the case in remote sensing?
>
> Yes, our approach is generic and we now show with a new experiment that it works very well out of the box for aerial and remote sensing imagery, see the new section in Appendix J.
>
> > Figure 2 legend can be improved by adding more details to it.
>
> We have revised Figure 2 and provided a more detailed description.

---

> > ### Comment · Reviewer_PzT7 · 2024-11-25
> > **Response to authors**
> >
> > Thank you for taking the time to answer to my concerns and appreciate the improvements done to the manuscript. I am keeping my score since I still believe there are limited difference caompared to Xun et at. 2024.

---

> > > ### Author Response · Authors · 2024-12-04
> > >
> > > Thank you for your feedback and for acknowledging the improvements that we made. While we are pleased to have set a new state-of-the-art for channel-invariant models on microscopy images with DINO-BoC, we emphasise that our primary contribution lies in the rigorous experimental challenge to the growing consensus that joint-channel-encoding is indispensable beneficial in channel-heterogeneous domains. We kindly invite you to have a look at the general response where we discuss this in more detail.
> > >
> > > We believe our critical examination is essential to the scientific discourse, as it refines the understanding of when and why certain approaches succeed or fail, moreover it provides practical guidance for practitioners.

---

### Official Review · Reviewer_aZkt · 2024-11-04

**Soundness:** 3
**Presentation:** 3
**Contribution:** 1
**Rating:** 3
**Confidence:** 3

**Summary:**

This paper proposes solutions for the challenge of handling varying number of channels, in particular for microscopy images where the number of channels can be different in each dataset. The authors use three main self-supervised approaches on DINOv2, with Bag of Channel approach performing the best across the benchamrks used.

**Strengths:**

- The problem statement and related work are well-written, and the problem is relevant
- The experimental section is reach. The authors provide results and comparisons on several datasets and benchmarks
- The code is going to be publicly released for future uses.

**Weaknesses:**

The main weakness of the paper is the limited contributions. The best performing method, which the authors call Bag of Channels, was proposed by (Xun et al, 2024). This paper only applies this method on DINOv2. Although the experiments and comparisons, alongside introduction of the Hierarchical Attention Model is valuable, these contributions could be a better fit for other conferences/journals rather than ICLR.

**Questions:**

The experiments are performed on datasets with 2-5 channels,
- Is there any correlation between the number of channels and the method performances?
- Why didn't you use datasets with more channels (as stated in L38, up to 8 channels)?

---

> ### Author Response · Authors · 2024-11-24
>
> Thank you for acknowledging positive aspects of our work (well written, with a rich experimental section, usefulness of the work). Your primary concern, regarding the significance of our contribution, was shared by other reviewers, and we hence describe our clarifications, revisions, and additional experiments in detail in the general response section above.
>
> Addressing your concerns specifically, compared to Microsnoop (Xun et al 2024), DINO Bag of Channels (BoC) is different in both architecture (convolutional U-Net vs ViT) and SSL strategy  (Masked Auto-Encoder (MAE) vs DINOv2). Our main contribution is that, contrary to recent works that argued in favor join encoding strategies (Bourriez et al 2024, Bao et al 2024, Kraus et al 2024, Pham & Plummer 2024), we find that (surprisingly) at scale the winning channel-invariant strategy is to independently encode the channels. We think this is rather striking result given that (a) Xun et al 2024, in their Fig.S1, found that ViTs underperformed CNNs when training on single channels (BoC approach) and (b) Bouriez et al 2024, in their Fig.4, reported that the BoC approach in turn underperforms channel-adaptive ViTs.
>
> We further show that DINO substantially outperforms MAE when combined with ViTs and BoC (this result is now more prominently featured in a new Table-1). Finally, we report new results comparing DINO BoC to Microsnoop on the Cyclops dataset, which is highly out of domain in our case (images of yeast with unseen channels). We find that DINO BoC outperforms Microsnoop by a large margin, especially on rare classes (up to 30 points).
>
> As such, we report with DINO BoC a specific recipe that, unexpectedly, outperforms all other methods, and thus provides an (we think) important challenge to a swath of recent papers reporting the superiority of joint channel-encoding, published at venues like CVPR and ICLR.
>
> > Why didn't you use datasets with more channels (as stated in L38, up to 8 channels)?
>
> This is a great prompt: a key differentiator between methods that seek to reconcile the joint modeling of channels (to learn interactions) with varying number and content of input channels, such as Channel-ViT, and our method, is that the former is only able to adapt to varying number of input channels up to some maximum (see L235-237), whereas DINO-BoC is agnostic with respect to the number and kind of input channels. As such, a Channel-ViT trained on the 5 input channels of ExtendedCHAMMI cannot process e.g. the full JUMP-CP  8-channel dataset (which includes 3 extra brightfield channels). In contrast, we now include additional results (Appendix I) that show that in the same setting, DINO-BoC is not only compatible with the additional input channels, but productively leverages them to improve performance on our JUMP-CP benchmark. DINO-BoC thus outperforms Channel-ViT in generalization to unseen channel combinations both quantitatively (where the comparison is possible) and categorically.
>
> > The experiments are performed on datasets with 2-5 channels, is there any correlation between the number of channels and the method performances?
>
> We find that using more channels does not always lead to improved performance, as shown in (new) experiments on aerial imaging data (Appendix J), using four NAIP channels alone results in higher performance than when additional channels from Sentinel-2 are also used. On the other hand, adding the 3 additional brightfield images on top of 5 fluorescent channels in JUMP-CP improves performance on evaluation (see above). The value of additional channels likely depends on how well their semantics can be interpreted by the feature extractor, and their relevance to a given downstream task.

---

> > ### Comment · Reviewer_aZkt · 2024-12-02
> > **Response to authors**
> >
> > I would like to thank the authors for their rebuttal. After reading their responses and the discussions with other reviewers, my main concern is still unsolved. The paper is not providing novel contributions compared to Xun et at. 2024. Applying an existing method on a different architecture and SSL strategy is an incremental contribution. Also, I agree with Reviewer TY5V regarding the redundancy of the HA method in this paper. Overall, although I appreciate the authors' efforts for the rebuttal and their additional experiments, I am keeping my original score.

---

> > > ### Author Response · Authors · 2024-12-04
> > >
> > > Thank you for your feedback. While we are pleased to have set a new state-of-the-art for channel-invariant models on microscopy images with DINO-BoC, we emphasise that our primary contribution lies in the rigorous experimental challenge to the growing consensus that joint-channel-encoding is indispensable beneficial in channel-heterogeneous domains. We kindly invite you to have a look at the general response where we discuss this in more detail.
> > >
> > > We believe our critical examination is essential to the scientific discourse, as it refines the understanding of when and why certain approaches succeed or fail, moreover it provides practical guidance for practitioners.

---

### Author Response · Authors · 2024-11-24

We thank all five reviewers for their constructive feedback. In response, we provide new experimental results as well as broad revisions that, we think, substantially strengthen the paper. We here address the shared major concerns regarding the generality and novelty of our method. We also address more specific critiques for each reviewer individually.

**Domain generality**

A shared critique concerned the generality of our approach, which our title seemed to allude to, but which indeed was not experimentally documented in our results. To address this, we provide a new set of experiments on a new domain, specifically aerial images of a methane source prediction multi-sensor dataset, Meter-ML, demonstrating that DINO-BoC here too achieves performance on par with the SOTA while using far less pre-training data and frozen features (Appendix J). We also narrowed our title to “Scaling Channel-Invariant Self-Supervised Learning for Microscopy Images”.

| Approach                                    | Test dataset | Pre-training dataset            | mAP   |
|---------------------------------------------|--------------|---------------------------------|-------|
| Meter-ML (Zhu et al 2022)               | NAIP, S2, S1 | NAIP, S2, S1 (85K)              | 51.7  |
| LHRS-bot (Muhtar et al 2024)              | NAIP, S2, S1 | LHRS-Align-Recap (1.1M images) | 71.8 |
| VHM	(Pang et al 2024) | NAIP, S2, S1 | VersaD (14M images) | 72.7 |
| DINO BoC (ours)                                | NAIP, S2     | Meter-ML NAIP, S2 train (85K)   | 70.9  |

| Approach                                    | Test dataset | Pre-training dataset            | mAP   |
|---------------------------------------------|--------------|---------------------------------|-------|
| Meter-ML (Zhu et al 2022)               | NAIP         | NAIP (85K)                      | 54.8  |
| SatMAE (Cong et al 2022)                | NAIP         | fMoW Sentinel                   | 76.9  |
| ScaleMAE (Reed et al 2023)               | NAIP         | fMoW RGB (363K)                 | 78.4  |
| USatMAE (Irvin et al 2023)                | NAIP         | USAtlas NAIP (3.6M)             | 83.7  |
| DINO BoC (ours)                                | NAIP         | NAIP  (85K)                     | 82.2  |
| DINO BoC (ours)                                | NAIP         | NAIP, S2 (85K)                  | 81.9  |

---

> ### Author Response · Authors · 2024-11-24
>
> Continuation:
>
> **Novelty**
>
> A primary concern shared by all reviewers is how our proposed method is different from previous works. We have thus substantially revised the narrative, presentation of our results, and provide additional experiments to more clearly delineate our work. Briefly, encoding channels independently, as proposed by Xun et al 2024 initially showed promise. However, rather than sacrificing the ability to learn channel-interactions, a recent set of high-profile works at CVPR (Kraus et al 2024, Bourriez et al 2024), ICLR (Bao et al 2024), and NeurIPs (Pham & Plummer 2024) reported considerable advantages for channel-adaptive methods that reconcile the need to account for variable channel number with the ability to joint encoding of channels (as is established practice for RGB images). Bourriez et al 2024 reported substantial gains of such methods compared to independent channel encoding (BoC approach). Xun et al 2024 further found that convolutional U-Nets outperformed ViTs in the BoC setting. Surprisingly, and contrary to this emerging consensus, we here show that, at scale, the specific combination of BoC with ViTs is the winning recipe. Moreover:
>
> In a revised Table-1, we highlight that DINOv2 substantially outperforms MAE, i.e. the SSL method used in Microsnoop, as a learning objective for BoC, and does so even when handicapped by a ViT-S instead of ViT-L. We note that even with MAE, BoC still outperforms Channel-ViT on 3 of 4 benchmarks. This indicates that both the ViT architecture, and the DINOv2 as the SSL objective (as the key technical differences to Microsnoop), independently synergize with the BoC approach at scale.
>
> | **Model**      | **SSL method** | **Network size** | **Channel invariant** | **Training set**    | **HPA-FOV F1 (Protein loc.)** | **HPA-FOV F1 (Cell type)** | **JUMP-CP Accuracy** | **WTC F1 (Cell cycle st.)** |
> |-----------------|----------------|------------------|------------------------|----------------------|-------------------------------|----------------------------|-----------------------|----------------------------|
> | **Channel-ViT** | DINOv2         | ViT-L            | ✓                      | ExtendedCHAMMI      | 56.7  | 90.4  | 39.5  | 87.2  |
> | **BoC**         | MAE            | ViT-L            | ✓                      | ExtendedCHAMMI      | 54.0 | 90.8  | 39.3  | 89.4  |
> | **BoC**         | DINOv2         | ViT-S            | ✓                      | ExtendedCHAMMI      | 55.6  | 90.7 | 44.5  | **91.0**  |
> | **BoC**         | DINOv2         | ViT-L            | ✓                      | ExtendedCHAMMI      | **61.7** | **91.1**  | **45.2**  | 90.5  |
>
> We further provide new results in Table-2, demonstrating that DINO BoC outperforms Microsnoop on the challenging Cyclops dataset (highly out of domain) by a large margin, and in particular for rare classes (by up to 30 points).
>
> | **Class**          | **Budtip** | **Cell periphery** | **Budneck** | **Actin** | **All** |
> |---------------------|------------|--------------------|-------------|-----------|---------|
> | **Frequency**       | 1.5%       | 1.9%               | 2.4%        | 3.8%      |         |
> | **Microsnoop** [Xun et al., 2024] | 32.1       | 96.4               | 43.4        | 48.0      | 75.9    |
> | **DINO BoC (Ours)**            | **62.1**   | **97.5**           | **72.6**    | **63.7**  | **83.1** |
>
>
> We further emphasize and expand our results on generalization to unseen channel combinations, as a key metric for the practical utility of foundation models in scientific domains with variable channel combinations. In particular, we show that DINO BoC not only outperforms Channel-ViT quantitatively, but in some instances categorically, since Channel-ViT cannot process datasets with a number of channels that exceeds that of its training dataset (Table-10).
>
> |                  | **JUMP-CP 5 channels** | **JUMP-CP 8 channels** |
> |------------------|-------------------------|-------------------------|
> | **Channel ViT**  | 39.5                   | ✗                       |
> | **DINO HA (Ours)** | **45.2**               | 51.4                    |
> | **DINO BoC (Ours)**         | **45.2**               | **51.6**                |
>
> Ideally, we would also compare the Microsnoop implementation (U-Net + MAE) trained on ExtendedCHAMMI directly to DINO BoC, but Xun et al did not release training code. Yet, even absent this comparison, our results provide an important beacon to the field: our findings refute the claimed superiority of joint-channel modeling channel-invariant learning at scale. By identifying surprising synergies between DINO, ViTs and BoC, we publish the most powerful channel-invariant model for microscopy to date, compatible with any number of input channels. We open-source both weights and all code for further improvement by the community.

---

> > ### Author Response · Authors · 2024-11-24
> >
> > Continuation:
> >
> > **Self-Supervised Learning (SSL) strategy**
> >
> > A recurring question has been about the choice of DINOv2 as the SSL strategy. While a comparison of different SSL strategies was made in Section 4.5 of the original manuscript, it is a pertinent comment that has led us to feature this discussion more prominently on the paper.
> >
> > In Table-1 we compare the performance of Bag of Channels (BoC) pre-trained with DINO -- a contrastive SSL method -- to MAE -- a masked SSL strategy. The results show that that the SSL strategy plays a significant role in the performance of BoC:
> >
> > | Pretraining strategy | Network size | HPA-FOV F1 Protein loc. | HPA-FOV F1 Cell type | Accuracy on JUMP-CP | WTC F1 Cell cycle st. |
> >  |-----------------------|--------------|--------------------------|----------------------|---------------------|-----------------------|
> > | MAE | ViT-L | 54.0 | 90.8 | 39.3 | 89.4 |
> > | DINO BoC | ViT-S | 55.6 | 91.0 | 44.5 | 91.0 |
> > | DINO BoC | ViT-L | 61.1 | 91.2 | 44.9 | 91.0 |
> >
> > In general, there is no obstacle to combining any SSL framework with the strategy of independently encoding each channel. Our finding was that the combination of BoC, DINO and ViTs outperforms all channel-invariant models to date.

---

### Author Response · Authors · 2024-12-04

We thank the reviewers for taking the time to consider our answers and for their feedback. We would like to emphasize that the significance of our work lies in the finding that a technically simple strategy, that of encoding each channel independently, when combined with ViTs and the DINOv2 SSL method (DINO BoC model), outperforms all previously proposed channel-invariant models for microscopy. This includes those that incorporate more complex joint-channel-encoding strategies. As a result, we not only set a new state-of-the-art for microscopy images but, more importantly, put forth a new direction for the field and provide critical guidance for practitioners.

Recent papers on channel-invariant methods (Kraus et al 2024, Bourriez et al 2024, Bao et al 2024, Pham & Plummer 2024) have built a consensus suggesting that joint-channel-encoding is indispensable in the channel-heterogeneous settings. Our experimental results, however,  demonstrate for the first time that joint-channel-encoding can, in fact, be detrimental.

We support this claim with a substantial set of experimental results:
- (Section 4.3) DINO BoC outperforms all other channel-invariant models when trained on a mix of microscopy datasets and evaluated both on known and novel data sources.
- (Section 4.4) DINO BoC consistently surpasses joint-channel-encoding methods on cross-dataset generalization tasks.
- (Section 4.5) DINO BoC establishes a new state-of-the-art on the CHAMMI benchmark (Chen et al 2024), a challenging benchmark for OOD generalization tasks. To the best of our knowledge, CHAMMI is the only standardized microscopy benchmark for channel-invariant models.

Our claim is further substantiated by experiments on DINO HA -- a technically innovative method -- that despite outperforming all previous approaches, still falls behind DINO BoC. The experiments with DINO HA are significant because this approach balances the strategies of jointly and independently encoding the channels, therefore it solidifies our argument and understanding of why certain approaches succeed or fail.

**Paper structure**

In response to the reviewers’ constructive suggestions, we have significantly reorganised our paper to better present our results.

Methodological details are now more clearly outlined in Section 2 (Method). Section 4 (Experiments) has been restructured to better highlight experimental results that were previously not sufficiently emphasized and additional experiments were incorporated:
- The choice of the SSL method DINOv2 is explained in Section 4.3, where we show that DINOv2 yields significant improvements over MAE, as demonstrated in Table 1.
- We present novel experimental results directly comparing DINO BoC to Microsnoop. Notably, DINO BoC outperforms Microsnooop on the Cyclops dataset, where Microsnoop had reported its largest gains, despite being highly out-of-domain for DINO BoC.

---

> ### Author Response · Authors · 2024-12-04
>
> **Novel Results on Aerial Imaging**
>
> We also demonstrate that DINO BoC holds promise for imaging domains other than microscopy. Specifically, we train and benchmark the performance of DINO BoC and DINO Channel-ViT on the Meter-ML aerial imagery dataset.
>
> DINO BoC consistently outperforms DINO CHannel-ViT. Furthermore, using the exact same normalization and evaluation protocol as for the microscopy benchmarks, DINO BoC performs on par or surpasses state-of-the-art models for aerial images. This is achieved even though DINO BoC is pre-trained on a much smaller dataset and does not use remote sensing specific architectures. This highlights the generality of our approach across a wider range of imaging domains.
>
>
> | Approach                  | Architecture | Test Dataset   | Pre-training Dataset                               | mAP  |
> |---------------------------|--------------|----------------|---------------------------------------------------|-------|
> | Meter-ML [Zhu et al. 2022]| DenseNet-121 | NAIP, S2, S1   | NAIP, S2, S1 (85K)                                | 51.7  |
> | LHRS-bot [Muhtar et al. 2024] | VLM          | NAIP, S2, S1   | LHRS-Align-Recap (1.1M images & text)            | 71.8  |
> | VHM [Pang et al. 2024]    | VLM          | NAIP, S2, S1   | VersaD (14M images & text)                       | 72.7  |
> | DINO Channel ViT          | ViT-L        | NAIP, S1, S2   | Meter-ML NAIP, S1, S2 train (85K)                | 67.5  |
> | DINO BoC                  | ViT-L        | NAIP, S1, S2   | Meter-ML NAIP, S1, S2 train (85K)                | **76.6** |
>
>
>
> | Approach                  | Architecture | Test Dataset   | Pre-training Dataset                               | mAP  |
> |---------------------------|--------------|----------------|---------------------------------------------------|-------|
> | Meter-ML [Zhu et al. 2022]| DenseNet-121 | NAIP           | NAIP (85K)                                       | 54.8  |
> | Supervised [Cong et al. 2022] | ViT-L        | NAIP           | fMoW Sentinel (770K)                             | 69.7  |
> | SatMAE [Cong et al. 2022] | ViT-L        | NAIP           | fMoW Sentinel (770K)                             | 76.9  |
> | ScaleMAE [Reed et al. 2023] | ViT-L        | NAIP           | fMoW RGB (363K)                                  | 78.4  |
> | USatMAE [Irvin et al. 2023]| ViT-L        | NAIP           | USAtlas NAIP (3.6M)                              | **83.7** |
> | DINO Channel ViT          | ViT-L        | NAIP           | NAIP, S1, S2 (85K)                               | 60.7  |
> | DINO Channel ViT          | ViT-L        | NAIP           | NAIP (85K)                                       | 68.6  |
> | DINO BoC                | ViT-L        | NAIP           | NAIP, S1, S2 (85K)                               | 76.5  |
> | DINO BoC                 | ViT-L        | NAIP           | NAIP, S2 (85K)                                   | 81.9  |
> | DINO BoC                | ViT-L        | NAIP           | NAIP (85K)                                       | 82.2  |
>
>
> **Additional Experiments**
>
> We have also included additional results addressing individual concerns raised by the reviewers, such as an ablation study on the feature dimension of DINO BoC and DINO Channel-ViT on Appendix K, as well as results for tasks with 8 channels on Appendix I (previously the tasks were limited to 5 channels).
>
> All of those experiments reinforce our core conclusion: independent channel-encoding is the most robust channel-invariant strategy, outperforming all other methods, even those based on more sophisticated techniques tailored to the problem setting; the pervasive assumption that joint-channel-encoding is necessary in channel-heterogeneous domains does not hold, specially at scale. In summary, through a rigorous analysis of existing methods, we challenge the previously held understanding of channel-invariant learning, which we believe is vital for advancing scientific discourse.

---

### Meta-Review · Area_Chair_1sU9 · 2024-12-19

**Metareview:**

This paper investigates self-supervised learning regimes for heterogeneous multi-channel images. The submission uses DINOv2 as a backbone and integrates bag of channels and hierarchical attention for channel-variant images. Initial experimental work concerned microscopy images (latterly aerial images). Investigation of challenges, specific to channel-varying images, can be considered an interesting pursuit and work in this area is welcome.

Post-rebuttal, a clear majority of reviewers remain unconvinced regarding the technical innovation and opt to retain their overall negative view on the paper. AC does not find a clear motivation to overrule in this case and therefore recommends rejection.

Authors received extensive feedback relating to the technical aspects of the work and are encouraged to take this onboard towards successful resubmission at an alternative venue.

**Additional Comments On Reviewer Discussion:**

The paper received reviews from five reviewers: one borderline accept, two borderline rejects and two clear rejects. Multiple reviewers recognised the extensive experimental work towards helping empirical understanding. Some queries could be partially alleviated during the rebuttal period however there were core and common concerns related to the incremental nature of the technical contributions.

Post-rebuttal, a clear majority of reviewers remain unconvinced regarding the technical innovation and opt to retain their overall negative view on the paper.

---

### Decision · Program_Chairs · 2025-01-22

Reject